



# Probabilistic modelling of inherent field-level pesticide pollution risk in a small drinking water catchment using spatial Bayesian Belief Networks

Mads Troldborg[1], Zisis Gagkas[1], Andy Vinten[1], Allan Lilly[1], and Miriam Glendell[1]

[1]The James Hutton Institute, Craigiebuckler, Aberdeen, AB15 8QH, Scotland, UK

*Correspondence to*: Mads Troldborg (mads.troldborg@hutton.ac.uk)

**Abstract.** Pesticides are contaminants of priority concern that continue to present a significant risk to drinking water quality. While pollution mitigation in catchment systems is considered a cost-effective alternative to costly drinking water treatment, the effectiveness of pollution mitigation measures is uncertain and needs to be able to consider local biophysical, agronomic,

and social aspects. We developed a probabilistic decision support tool (DST) based on spatial Bayesian Belief Networks (BBN) that simulates inherent pesticide leaching risk to ground- and surface water quality to inform field-level pesticide mitigation strategies in a small drinking water catchment (3.1 km$^2$) with limited observational data. The DST accounts for the spatial heterogeneity in soil properties, topographic connectivity, and agronomic practices; temporal variability of climatic and hydrological processes as well as uncertainties related to pesticide properties and the effectiveness of management

interventions. The rate of pesticide loss via overland flow and leaching to groundwater and the resulting risk of exceeding a regulatory threshold for drinking water was simulated for five active ingredients. Risk factors included climate and hydrology (temperature, rainfall, evapotranspiration, overland and subsurface flow), soil properties (texture, organic matter content, hydrological properties), topography (slope, distance to surface water/depth to groundwater), land cover and agronomic practices, pesticide properties and usage. The effectiveness of mitigation measures such as delayed timing of pesticide

application; 10%, 25% and 50% reduction in application rate; field buffers; and presence/absence of soil pan on risk reduction were evaluated. Sensitivity analysis identified the month of application, land use, presence of buffers, field slope and distance as the most important risk factors, alongside several additional influential variables. Pesticide pollution risk from surface water runoff showed clear spatial variability across the study catchment, while groundwater leaching risk was uniformly low, with the exception of prosulfocarb. Combined interventions of 50% reduced pesticide application rate, management of plough pan,

delayed application timing and field buffer installation notably reduced the probability of high-risk from overland runoff and groundwater leaching, with individual measures having a smaller impact. The graphical nature of the BBN facilitated interactive model development and evaluation with stakeholders to build model credibility, while the ability to integrate diverse data sources allowed a dynamic field-scale assessment of 'critical source areas' of pesticide pollution in time and space in a data scarce catchment, with explicit representation of uncertainties.



## 1 Introduction

Diffuse pesticide pollution continues to represent a significant risk to surface and drinking water quality worldwide (Villamizar et al., 2020). European Union legislations (Water Framework Directive (WFD) (European Commission, 2000), and the related Drinking Water Directive (DWD) (European Commission, 1998), and Groundwater Directive (European Commission, 2006)) require that concentration of individual pesticides in drinking water must not exceed 0.1 µg L$^{-1}$ and the total concentration of

all pesticides must be below 0.5 µg L$^{-1}$. Article 7 of the WFD promotes a 'prevention-led' approach to DWD compliance, prioritising pollution prevention at source rather than costly drinking water treatment. Catchment management schemes are therefore now widely adopted by policy makers and water companies to mitigate diffuse pollution by pesticides (and other pollutants) and to improve the raw water quality prior to treatment. However, the effectiveness of such diffuse pollution mitigation measures is uncertain due to the heterogeneous nature of catchment systems, and hence catchment management

needs to be targeted to consider local biophysical, agronomic, and social aspects (Okumah et al., 2018). To select and prioritise cost-effective interventions, it is essential to identify and map 'high risk' areas, often referred to as critical source areas (CSAs), i.e. those areas within a catchment that contribute disproportionately large amounts of pollutants to a given water quality problem (Doody et al., 2012; Reaney et al., 2019).

Modelling approaches are commonly used to identify diffuse pesticide pollution risk areas and to help evaluate the effectiveness of mitigation strategies. While process-based distributed models, such as the Soil and Water Assessment Tool (SWAT), have been widely used to simulate transport, fate and risks of pesticides at catchment scale and to evaluate the effectiveness of interventions (Babaei et al., 2019; Villamizar et al., 2020; Wang et al., 2019), their application is computationally costly and often hindered by lack of monitoring data for model calibration and validation. Therefore, various

spatial index models have been developed to evaluate the intrinsic vulnerability and risk from pesticide pollution at a range of scales (Kookana et al., 2005; Stenemo et al., 2007; Worrall and Kolpin, 2003). The simplest index-based methods assign scores and weights to a set of spatially distributed indicators (e.g., soil media, recharge rate, and depth to groundwater and contaminant properties), which are then combined into an overall risk index, typically within a GIS environment. An example of such index-based method is the DRASTIC system (Aller et al., 1985), which has been widely used for groundwater vulnerability

mapping and for identifying areas most at risk to pollutant leaching. DRASTIC only considers geological and hydrogeological factors but ignores the specific nature of the contaminant(s), and it is therefore classed as an intrinsic vulnerability method. Several modifications and methods similar to DRASTIC exist in the literature, many of which aim to provide specific vulnerability maps, where the contaminant source and behaviour are also accounted for (Duttagupta et al., 2020; Nobre et al., 2007; Saha and Alam, 2014). Other studies have used simplified 1D solute transport models to develop indices and rankings

of potential pesticide leaching (Gustafson, 1989; Jury et al., 1987; Stenemo et al., 2007), while other methods such as the SCIMAP modelling framework (Lane et al., 2009; Reaney et al., 2011) use digital elevation models to derive spatial patterns of relative potential erosion and hydrological connectivity to identify possible critical source areas for diffuse pollution risk.





While index-based vulnerability methods are useful for initial screening purposes, they also have several limitations. Index-
based methods do not account for uncertainties in model parameters and complex processes, and they lack probabilistic
integration of lines of evidence (Carriger et al., 2016; Carriger and Newman, 2012). In addition, the scores and weights are
typically assigned subjectively, and different scoring-systems can therefore provide substantially different results. Finally,
index-based methods usually do not account for actual concentration data, and poor correlation between vulnerable areas and
field concentration measurements have been reported (Worrall and Kolpin, 2003).


To address the first two shortcoming, we developed a probabilistic decision support tool (DST) using spatial Bayesian Belief
Networks (BBN) to inform field-level pesticide mitigation strategies in a small drinking water catchment (3.5 km$^2$) with limited
observational data. BBNs are probabilistic graph-based models that allow to integrate various information sources, including
different types of data, literature and expert opinion into a single modelling framework, thus maximising the use of both
available knowledge and data (Carriger et al., 2016; Carriger and Newman, 2012). In BBNs, model variables and their causal
relationships are represented as 'nodes' and 'arcs' in a so-called Directed Acyclic Graph (DAG) (i.e., a graph that has no
feedback loops). The graphical nature of a BBN lends itself to collaborative model co-construction with experts and
stakeholders and helps to build model credibility. A major advantage of the BBN approach is that it allows to carry out
probabilistic inference based on (uncertain) evidence. Probabilistic inference is simply the task of calculating the posterior
probability distribution of the BBN given the available observations and can be both predictive (i.e., reasoning from new
observations of causes to new beliefs about the effects) and diagnostic (i.e., reasoning from observed effects to updated beliefs
about causes).

The use of BBNs has gained increasing popularity in environmental modelling and risk assessment (Aguilera et al., 2011;
Kaikkonen et al., 2021) with examples including pesticide risk management (Carriger and Newman, 2012; Henriksen et al.,
2007) and probabilistic assessments of pesticide exposure and effects (Mentzel et al., 2021). While the integration of Bayesian
networks with GIS in environmental risk assessment has also been growing steadily over recent years (Moe et al., 2021), to
date spatial BBN has only been used for pesticide risk modelling on a single occasion  to assess pesticide runoff risk at a basin
scale across France (Piffady et al., 2020). To our knowledge, the present study provides a first application of spatial BBN for
probabilistic field-level assessment of intrinsic pesticide pollution risk from 'critical source areas' at a monthly resolution.

The aim of this research was to examine the spatial variability of risk factors within an uncertainty framework to better inform
field-level targeting of management interventions in a small drinking water catchment with limited available data. Specifically,
we sought to answer the following questions: a) Can we characterise the spatial and temporal variability of pesticide pollution
risk from groundwater leaching and overland flow in a data-sparce catchment? b) Which factors are most influential on intrinsic
pesticide pollution risk? c) What is the effectiveness of available management interventions on pesticide risk reduction?


## 2 Methods

### 2.1 Study site

Jersey Island (c. 117 km$^2$) (49.2138 °N, 2.1358 °W), the largest in the English Channel group, comprises a plateau with an
elevation 60-120 m above sea level (Robins and Smedley, 1998). The climate on Jersey is oceanic with average temperature
ranging from 6 °C in winter to 18 °C in summer and mean annual rainfall around 900 mm. A shallow, fractured bedrock aquifer
underlies most of the island with a generally shallow depth to the water table increasing to 10-30 m beneath higher ground.
For the most part, groundwater storage and transport is shallow and within the top 25 m of the saturated rock (Robins and
Smedley, 1998). Bedrock is Precambrian to Cambro-Ordovician. The west-central part of the island is mostly underlain by the
oldest rocks belonging to the Jersey Shale Formation, while a volcanic formation occupies much of the east. Superficial deposit
include wind-blown sand, loess and alluvium (Robins and Smedley, 1991).

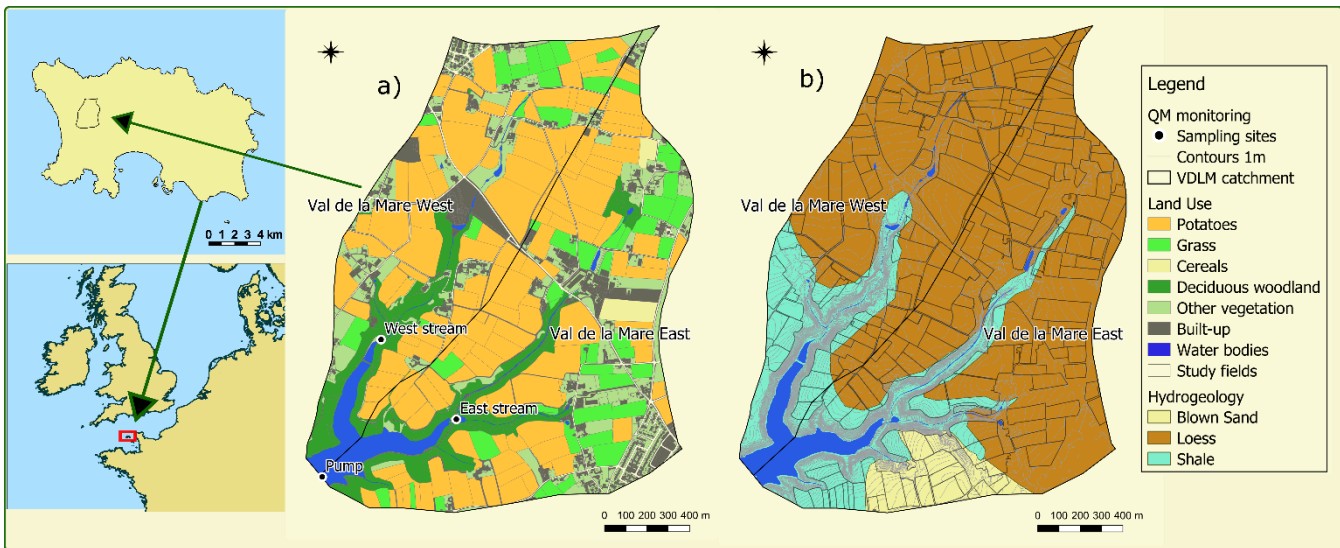

**Figure 1: Location of the Val de la Mare (VDLM) study catchment on the Island of Jersey. Fig 1a shows the three water quality**
**monitoring sites (Pump, West stream, and East stream) and the land use in the catchment, while Fig 1b shows the hydrogeology.**
**Country boundaries were taken from open source www.gadm.org and hydrogeological data was digitised from the Hydrogeological**
**Map of Jersey under Open Government Licence v3.0 (Contains British Geological Survey materials © UKRI 1992)**

Historically, water resources across the Island of Jersey have been vulnerable to nitrate and pesticide pollution, particularly
during the growing season when concentrations in untreated water can exceed regulatory drinking water quality levels. This
can adversely affect the raw water quality within impounding reservoirs, requiring the water company to undertake a series of
mitigation measures at the treatment works to avoid breaches in the treated water supply. The small size of the Island means
that land-use across the island is dominated by intensive agriculture, primarily potato and dairy farming. The cultivation of the
Jersey Royal Potato takes place during the growing season from January to May. There is very little crop rotation and

accordingly the potato crop is grown with the support of man-made fertilisers and pesticides (herbicides, fungicides and nematicides).

This study focused on the Val de la Mare (VDLM) catchment (3.1 km$^2$) in south-west of Jersey, which feeds into the VDLM Reservoir, the second largest reservoir in Jersey constructed in 1962. The reservoir holds up to 938,700 m$^3$ of untreated water,

enough to supply Jersey with water for approximately five weeks. Water feeds into the reservoir from within the catchment area, as well as from neighbouring catchments and a desalination plant when it is in operation. The water quality in the VDLM reservoir is vulnerable to pesticide pollution, with pesticide concentrations often exceeding the regulatory drinking water quality levels of 0.1 µg l$^{-1}$ for individual pesticides and 0.5 µg l$^{-1}$ for total pesticide concentration (European Commission, 1998).


To evaluate the spatial and temporal variability of the intrinsic pesticide pollution risk from critical source areas within the VDLM study catchment, we have developed a probabilistic model based on spatial Bayesian Belief Networks (BBN). The data and information used to inform the model development are presented in the following sections (2.2 - 2.4), while the BBN model itself is described in section 2.5. For the purpose of this paper, we define risk as the probability that the pesticide

exposure (i.e., the pesticide fluxes from the fields to the reservoir) results in the regulatory drinking water standards to be exceeded.

## 2.2 Pesticide detection, usage and properties

Five active pesticide ingredients currently or recently in use in the catchment showed evidence of significant concentrations in the reservoir offtake for the drinking water supply. These included the herbicides glyphosate, metobromuron, pendimethalin

and prosulfocarb, and the nematicide and insecticide ethoprophos. Metobromuron was most frequently observed above the drinking water standard, followed by ethoprophos, prosulfocarb and pendimethalin (Table 1). Ethoprophos was not included in the final model, as its use has now been discontinued, instead the nematicide fluopyram was considered as it can be a potential replacement for ethoprophos (Table 2). Fluopyram is used at lower application rates than ethoprophos, making the risk of contaminating the water supply intrinsically lower, notwithstanding its relatively high mobility and greater persistence.


Data on pesticide application rates and timing for 2016-2018 was obtained from the Jersey Royals Company, the main potato crop grower in the area who manage ca. 50% of the catchment area, alongside agronomic data on crop coverage and crop rotation for the 2010-2018 period. Pesticide usage (Table 2) was estimated assuming that the available agrochemical data was representative of the whole catchment. Glyphosate was mostly applied in January (mean day of application 23$^{rd}$ January),

while other pesticides were typically applied in February (mean day of application 11$^{th}$ to 16$^{th}$ February). Hence, only months January – March were represented in the model, to allow for a potential one-month delay in pesticide application. Usage of





pesticides on other crops in the catchment was limited to the use of glyphosate for spraying off grass prior to cultivation and use of pendimethalin on barley.

**Table 1: Summary of the pesticide monitoring data by location (P: Pump; E: East stream; W: West stream) in the VDLM catchment 2016-2019. Detection data are summarised as total number of samples, number of samples above limit of detection (LOD), and number of samples above the drinking water standard of 0.1 µg l⁻¹. The concentration data are summarised as mean, minimum and maximum observed concentration.**

| | Detection (total/ >LOD/ > 0.1 µg l⁻¹) | | | Concentrations (µg l⁻¹) (mean (min-max)) | | |
|---|---|---|---|---|---|---|
| **Pesticide** | **P** | **E** | **W** | **P** | **E** | **W** |
| Glyphosate | 34/33/0 | 20/20/0 | 21/21/0 | 0.033 (0.004-0.083) | 0.031 (0.005-0.093) | 0.029 (0.006-0.10) |
| Metobromuron | 27/27/27 | 4/4/4 | 6/6/6 | 0.223 (0.1-0.4) | 0.2 (0.1-0.40) | 0.25 (0.2-1.7) |
| Pendimethalin | 258/107/0 | 129/42/0 | 245/122/5 | 0.01 (<0.005-0.07) | 0.006 (<0.005-0.02) | 0.017 (<0.005-0.28) |
| Prosulfocarb | 67/55/14 | 6/3/1 | 6/6/3 | 0.098 (<0.002-1.01) | 0.048 (<0.002-0.26) | 0.318 (0.001-1.25) |
| Ethoprophos | 181/137/15 | 105/36/5 | 101/56/11 | 0.033 (<0.002-0.27) | 0.015 (<0.002-0.24) | 0.073 (<0.002-2.43) |

Key pesticide properties for assessing the risk to water quality were extracted from a publicly available database (Lewis et al., 2016) (Table 2). These included the $K_{oc}$ coefficient that represents the adsorption of the pesticide onto the organic carbon of soil, subsoil and vadose zone materials and the half-life (DT50) that represents the degradation rates during transport through each of these layers (Table 2). A third process, that of volatilisation was not considered as this is relatively minor in most cases and omission of this process will provide a conservative estimate for impact to groundwater. Retention of pesticides by the

soil and subsoil materials to which it is applied depends on adsorption to organic and mineral surfaces. Soil sorption processes are complex and have been the subject of substantial research in the past. Pesticide sorption is influenced by both the chemical characteristics of the pesticide and soil specific properties, such as soil organic carbon (SOC) concentration, clay content, pH and soil moisture content. For neutral non-polar pesticides, it is well documented that pesticide retention is strongly correlated to SOC concentration, with other factors such as clay content, pH and aeration status playing a subsidiary role (Kah and Brown,

2006; Wauchope et al., 2002). For weakly ionisable pesticides with ionic equilibrium constants near the range of soil pH, sorption may be highly sensitive to the pH of the sorbing soil. However, none of the pesticides considered for the modelling here are considered weakly ionisable, and therefore only the role of SOC content on pesticide retention was considered in the model by including the pesticide adsorption coefficient $K_{oc}$. The degradation rate of pesticides in soil and subsoil depends principally on the microbiologically-mediated decomposition and as such is also strongly influenced by SOC (Jury et al., 1987;

Kah and Brown, 2006). Some chemical degradation of pesticides on mineral surfaces can also occur, which may be more important for the subsoil and vadose zone, but in most cases, biological degradation is seen as the main pathway (Fomsgaard, 1995). In most cases, the time for 50% disappearance (DT50) or 90% disappearance (DT90) is determined.





**Table 2 Summary of pesticide properties considered in the model (Lewis et al. 2016) and mean application rates in the study**
**catchment. AI=Active Ingredient. $K_{oc}$ = pesticide adsorption coefficient on soil organic carbon. DT50 = time for 50% of pesticide**
**to be degraded in soil.**

| | Type of pesticide | Typical values | | Application rates kg AI ha$^{-1}$ | Typical application timing |
|---|---|---|---|---|---|
| | | $K_{oc}$ ml g$^{-1}$ | DT50 (field) days | | |
| Glyphosate | Herbicide | 1424 (884-50660) | 23.8 (5.7-40.9) | 0.9 | January |
| Metobromuron | Herbicide | 197 (122-199) | 22.4 (5.4-64.5) | 1.25 | February |
| Pendimethalin | Herbicide | 17491 (10241-36604) | 100.6 (39.8-187) | 0.9 | February |
| Prosulfocarb | Herbicide | 1693 (1367-2339) | 9.8 (6.5-13) | 3.2 | February |
| Fluopyram | Fungicide/ Nematicide | 279 (233-400) | 118.8 (93.2-144.6) | 0.25 | February |

## 2.3 Catchment characteristics

Monthly total rainfall and mean monthly temperature data (from 1894-2019) were obtained from the Government of Jersey
website (https://opendata.gov.je/organization/weather). Mean monthly potential evapotranspiration (PET) and actual
evapotranspiration (AET) were calculated using the approach in (Pistocchi et al., 2006), see Appendix A.

Spatial environmental data were processed and collated in a single GIS shape file. Visualisation, geographical analysis, and
processing of spatial datasets were done in QGIS 3.12.2 (QGIS.org, 2021. QGIS Geographic Information System. QGIS
Association. http://www.qgis.org), while ArcGIS 10.1 (ESRI 2011. ArcGIS Desktop: Release 10. Redlands, CA:
Environmental Systems Research Institute) was used for the generation of the digital terrain model. The following data sets
were used to inform model parameterisation.

Land parcels that fell within the VDLM catchment area were selected and filtered using Feature types (i.e. cultivation) to
identify cultivated fields. The field selection was supplemented by visual inspection using satellite imagery (Google Satellite
service) to ensure that only cultivated fields were selected. This resulted in the selection of 200 fields, which were assigned a
dominant crop type for the 2010-2018 period by spatially joining field polygons with layers containing crop type information.
Crop operation information available for 56 fields within the catchment was used to determine the timing and pesticide
application rates (Table 2) and to inform the prior distributions in the model.

A 1m resolution hydrologically-corrected DTM of the VDLM catchment area was created using digital line contours at 1 meter
interval provided by Jersey Water and the 'Topo to Raster' tool in ArcGIS; the DTM was used to calculate mean elevation and





slope (in degrees) for each field polygon within the VDLM catchment. Topographic connectivity was derived by calculating the horizontal distance from the polygon vertex nearest to the stream features using the Distance to nearest hub tool in QGIS.


The overall depth of the soil column is a key characteristic of the soil that influences the attenuation of surface applied pesticides. Depth to groundwater was calculated for each field using a 5 m groundwater level contour map of Jersey prepared by the British Geological Survey and provided by the Jersey Water company (T. de Feu, pers. comm, May 20, 2020) by importing and georeferencing the map in QGIS, digitising the contour lines and combining them with field polygons.


Soil water retention, conductivity, natural drainage, depth to groundwater and anthropogenic characteristics such as plough pans were considered in model parameterisation. There has been no systematic survey of the soils of Jersey in the traditional sense of classifying and grouping soils according to their pedology. Brief descriptions of 'soil series' were given in Jones et al. (1990) and the Soil Atlas of Europe (European Soil Bureau Network. Eds.: Jones et al. 2005) shows a single soil type

Dystric Cambisols for the whole Island of Jersey at a 1:2 500 000 scale. Due to this lack of detailed soil mapping, the hydrogeological map of Jersey (Robins et al., 1991) was used to identify soil hydrological units based on the three hydrogeological formations identified in the VDLM catchment (Fig. 1b). These three soil hydrological units consisted of soils developed on loess, aeolian (blown) sand and shales, respectively. Both loess and sand are periglacial Quaternary deposits that are relatively common throughout northern Europe. Soil hydrological data for similar soils is contained within the HYPRES

soil hydrological database (Wosten, et al. 1999; Wosten et al. 1998), hence this database was used to derive the soil hydrological properties necessary for the modelling of pesticide attenuation.

According to the descriptions of soil series in Jones et al. (1990), the soils developed on these three hydrogeological units are generally well to moderately well drained with no real inhibition to the downward movement of water. Soil hydrological

properties derived from HYPRES database supported this assumption, with mean saturated hydraulic conductivity largely greater than 10 cm day$^{-1}$. Less than this would indicate the presence of a slowly permeable horizon (MAFF, 1988) with some degree of ponding within the soil.

Local knowledge suggested that some of the soils in the catchment could have a plough pan potentially as deep as 60 cm below

the surface. This layer is likely to restrict the downward flow of water through the soil and increase the likelihood of near surface runoff and was included as one of the soil parameters that could be manipulated within the model. In the absence of data in the HYPRES database on plough pans, a value of 0.02 cm day$^{-1}$ was selected as a worst case estimate of plough pan hydraulic conductivity based on Koszinski et al. (1995) who reported hydraulic conductivity of compacted soil of between 0.02 and 3 cm day$^{-1}$.






Laboratory measurements of topsoil soil organic matter (SOM) content and pH were available for 40 and 37 fields within the VDLM catchment, respectively, most of which were cultivated with Jersey Royal potatoes (32). Soil organic matter (SOM) content ranged from 1.9% to 3.8% with mean SOM at 2.5%. Median SOM was slightly greater for the six fields found on grassland (3%) than in the potato fields (2.3%). Soil pH ranged from 5.3 to 7.2 with a mean pH value of 6.4 and mean soil pH

was slightly greater in the potato fields (6.4) than in grassland (6.1). However, as this information was not available for all fields and only for the topsoil, these data were not used in the model parameterisation. Instead, model sensitivity to using observed SOM concentrations vs. those derived from the HYPRES database was examined and was found to be negligible, therefore the converted HYPRES SOC values (by dividing SOM by 1.724) were used as priors in the model to ensure spatial consistency.


### 2.4 Field attributes

The spatial data described in 2.3 was used to inform the parameterisation of the model variables. It was found that most study fields within the VDLM catchment lay in relatively flat ground with 152 fields having a mean slope less than 3 degrees, while elevation range was also relatively small, from 67m to 99m. Loess was the dominant soil parent material in 155 fields, while

soils derived from Jersey Shale or Blown Sand were dominant in the remaining 31 and 14 fields, respectively. Loess covered the central and northern part of the VDLM catchment, while Shale was found in the south-western part and Blown Sand in the south-eastern part of the catchment. Most study fields had a sandy silt loam (120) or sandy loam (59) soil texture class. Fields with a sandy silt loam texture were mostly underlain by the Loess hydrogeological formation (101), while fields with sandy loam texture were mostly underlain by the Jersey Shale formation (36) and blown sand (12).


The horizontal distance of VDLM fields to the stream network was used to represent field connectivity to the stream network and the reservoir. The median horizontal distance was 119 m, with 91 fields located within 100 m of the stream network. Depth to groundwater within the VDLM catchment ranged from 0 to 20 m, but most of the catchment area (56%) had a shallow aquifer less than 5 m deep (117 fields), with further 54 fields having groundwater depths between 5 and 10m.


### 2.5 Spatial Bayesian belief network risk model

We have developed a probabilistic model, based on spatial Bayesian Belief Networks (BBN) (Appendix A), to evaluate the spatial and temporal variability of the intrinsic pesticide pollution risk from critical source areas within the VDLM study catchment. The model aims to provide a field level assessment of the relative water pollution risk characteristics of each field,

made available as probabilistic map layers. The developed approach integrates the various information sources described above and includes causal relationships between both discrete and continuous variables in a hybrid BBN. The general principles and theory of Bayesian networks have been described extensively elsewhere (Korb and Nicholson, 2010; Moe et al., 2021) and will not be discussed in detail here.





The model structure and development were informed by expert knowledge and stakeholder feedback. Hydrologically, the VDLM reservoir was assumed to be fed by the west and east streams (Fig. 1) as well as by groundwater, and the groundwater aquifer was assumed to be unconfined and homogenous. Thus, pesticides applied to a given field could either leach to groundwater or could be transported directly to the reservoir or one of the streams through surface runoff. The risk assessment model therefore accounted for pesticide losses via both groundwater leaching and overland flow, with the final assessment

based on the combination of both. Both the groundwater leaching and the overland flow, pesticide losses are influenced by three key factors, namely soil and site characteristics, climate and hydrology, and land management (e.g., pesticide usage and properties, land use) (Fig. 3, Appendix A).

    In the following, the modelling of the groundwater leaching and the overland flow risk components is described. In both cases,

we are interested in determining how much of the applied pesticide rate $L_0$ [M L$^{-2}$ T$^{-1}$] may eventually reach either groundwater via leaching ($L_{gw}$ [M L$^{-2}$ T$^{-1}$]) or surface water via overland flow ($L_{of}$ [M L$^{-2}$ T$^{-1}$]) from a given field. To ensure mass balance, we assumed that for each field only a proportion $f_{leach}$ of the applied pesticide would be available to leaching, while the remaining proportion would run off to surface water. During transport to groundwater and surface water the pesticide can undergo attenuation with the degree of attenuation determined by attenuation factors ($AF_{gw}$ and $AF_{of}$) (as described in the

following sections). Overall, the pesticide fluxes that reaches groundwater via leaching and surface water via runoff from a given field was therefore given by:

$$L_{gw} = L_0 * AF_{gw} * f_{leach} \qquad (1)$$

$L_{of} = L_0 * AF_{of} * (1 - f_{leach}) \qquad (2)$

    Conceptually, this way of calculating pollution risk to groundwater and surface water is similar to the pesticide impact rating index proposed by Kookana et al. (2005) and to the InVEST nutrient delivery model (Sharp et al., 2020). For the modelling here, it was assumed that the fraction of the applied pesticide to land that would be available to leaching $f_{leach}$ equalled the ratio

of infiltration to excess rainfall.

    The combined pesticide flux from a given field was the sum of the leaching and the overland flow component.
This combined pesticide flux was converted to a surface water concentration to evaluate the risk to surface water as follows:

$C_{sw} = (L_{gw} + L_{of}) * A_c / V_{res} \qquad (3)$





where $A_c$ [L²] is the total field area in the catchment (192 ha) and $V_{res}$ [L³] is the water volume in the reservoir (938,700 m³).
Eq. 3 is a very simplified way of converting the total pesticide flux from a given field to a concentration in the reservoir (it is
essentially assumed that the given field represents all fields in catchment), which allows a comparison to the regulatory

standards based on which the risk can subsequently be assessed. Hence, if the total pesticide flux from a field resulted in $C_{sw}$
exceeding the standard of 0.1 µg l⁻¹, this field was considered high risk (see Appendix A).

**2.5.1 Groundwater leaching risk assessment**

The conceptual model for the pesticide leaching to groundwater applied in this study catchment (Fig. 3) followed on from the

screening model proposed by Jury et al. (Jury, W.A., Dennis, D.F., Farmer, 1987). The model assumes that a single pesticide
mass flux is applied to the soil surface (z=0). The pesticide is assumed to dissolve in the infiltrating rainwater and move
downward through the soil profile by leaching at a constant infiltration rate $J_w$ [L T⁻¹], which is here determined by the amount
of excess rainfall and the physical properties of the soil (Appendix A). The pesticide is assumed to move downward through
the soil by piston flow (i.e. no dispersion) while undergoing linear adsorption and first-order decay. Given these assumptions,

the pesticide transport and fate can be described by the following mass balance equation (Jury et al. 1987):

$$RF \frac{\partial C}{\partial t} = -\frac{J_w}{\theta_w} \frac{\partial C}{\partial z} - \mu * RF * C \qquad (4)$$

where $C$ [M T⁻³] is the pesticide concentration in solution (the water phase), $\theta_w$ is the volumetric water of the soil, $\mu$ [T⁻¹] is

the first-order degradation constant, and $RF$ is the retardation factor given by (Jury et al. 1987):

$$RF = 1 + \frac{\rho_b * f_{oc} * K_{oc}}{\theta_w} \qquad (5)$$

where $K_{oc}$ [L³ M⁻¹] is the organic carbon partition coefficient, and $\rho_b$ [M L⁻³] and $f_{oc}$ [M M⁻¹] are the soil bulk density and the

organic carbon content of the soil, respectively. The retardation factor describes the velocity of the solute pesticide relative to
the infiltrating water. The solution to the mass balance equation (1) for an instantaneous mass injection $m_0$ can be written as:

$$m(z) = m_0 * AF(z) \qquad (6)$$

where m(z) [M L⁻²] is the pesticide mass contained within the pulse when it reaches depth z, and AF is the attenuation factor
given by (Stenemo et al., 2007):



$$AF(z) = \exp\left(-\frac{\mu * RF * \theta_w}{J_w} z\right) = \exp\left(-\frac{\ln(2) * RF * \theta_w}{J_w * DT50} * z\right) \qquad (7)$$

where DT50 [T] is the half-life of the pesticide. AF expresses the fraction of the applied pesticide that will reach depth z and
       can take values between 0 (none of the applied pesticide will reach depth z) and 1 (all the applied pesticide will reach depth
       z).

       It is well-known that organic matter concentration and microbial population density decrease with depth in soil profile, hence
both the pesticide retardation and decay rates are expected to decrease with depth (Jury et al., 1987; Kookana et al., 2005). To
       account for this effect, the model divided the soil profile into three regions (Fig. 2): a) the A horizon (topsoil), which was
       assumed to be 30 cm thick and to be the most microbially active region; b) the B horizon (subsoil), which was also assumed
       to be 30 cm thick; and c) the vadose zone, which extends from the bottom of the B horizon to the groundwater table and was
       assumed be microbially the least active. Furthermore, in the VDLM catchment, the presence of low-permeability soil pans was
believed to be widespread due to intensive management. When such soil pan is present, it was assumed to be 10 cm thick and
       to be situated within the B horizon (Fig. 2). Each of the regions were characterised by uniform values of volumetric water
       content, soil bulk density, organic carbon content and decay rates (Appendix A).

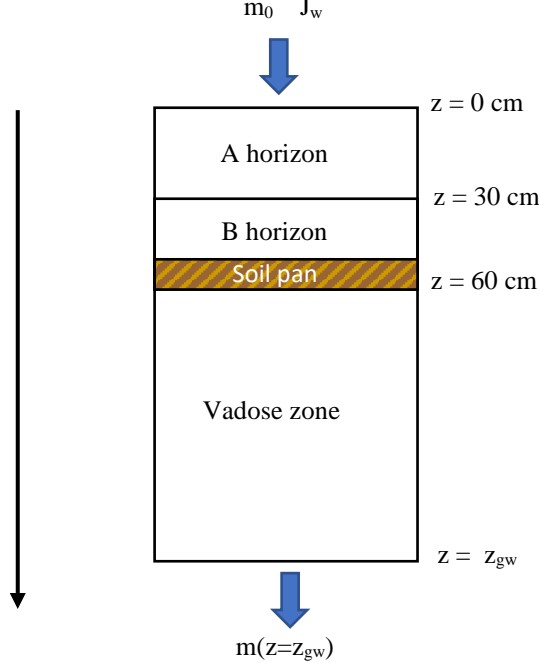

**Figure 2:**  **Conceptual model for pesticide leaching from soil surface to groundwater.**





An attenuation factor can be calculated for each of the zones:

$$AF_i = \exp\left(-\frac{\ln(2) * RF_i * \tau_{w\_i}}{DT50_i}\right) \qquad (8)$$

where $RF_i$ and $DT50_i$ are the retardation factor and half-life in zone $i$, and $\tau_{w\_i}$ [T] is the average time it takes the infiltrating water to travel through a given zone $i$:

$$\tau_{w\_i} = \frac{\theta_{w\_i} d_i}{J_w} \qquad (9)$$

where $\theta_{w\_i}$ and $d_i$ [L] are the effective volumetric water content and the thickness of horizon $i$, respectively.

The aim of the leaching risk model is to predict the pesticide flux reaching the groundwater $L_{gw}$ [M L$^{-2}$ T$^{-1}$] (Eq. 1). To do this, an effective attenuation factor $AF_{gw}$ is calculated by multiplying the AF for each of the three soil profile regions:

$$AF_{gw} = AF_A * AF_B * AF_{VZ} \qquad (10)$$

To assess whether the calculated pesticide flux that reaches groundwater is considered high, medium or low, the flux was converted to a pesticide concentration in groundwater $C_{gw}$ by assuming $L_{gw}$ was mixed in the upper $d_{mix} = 0.1$ m of the aquifer:

$$C_{gw} = L_{gw}/d_{mix} \qquad (11)$$

This predicted pesticide concentration in groundwater can be compared to measured concentrations (if available), detection limits and/or to regulatory standards. A pesticide flux to groundwater of 0.0001 kg ha$^{-1}$ yr$^{-1}$ or above was considered high, as this would result in the regulatory drinking water standard of 0.1 µg l$^{-1}$ to be exceeded if mixed in the top 0.1 m of groundwater,

whereas the threshold for low groundwater leaching fluxes was defined as being 10-fold less (Appendix A). Overall, the pesticide leaching risk assessment reflected a set of soil-specific factors (organic carbon concentration, bulk density, water content and thickness for each of the three layers), pesticide/management-specific factors (application rate, $K_{oc}$ and half-life), and climatic factors (rainfall and temperature from which groundwater recharge and runoff rates were estimated).

**2.5.2 Overland flow risk assessment**

The transport of pesticide from the site of application to surface water via overland flow is complex. Pesticides can leave a field either dissolved in runoff water or attached to eroded soil colloids. Because the amount of eroded soil lost from a field is





usually small compared with the runoff volume, losses via runoff are generally considered more important than losses via erosion for most pesticides. Only for strongly adsorbing pesticides, the erosion pathway becomes the more dominant
(Reichenberger et al., 2007). However, in the present study, potato cropping was the dominant land use, which has been shown to be highly erosive (Vinten et al., 2014), therefore the runoff and erosion transport pathways have been lumped into one for the modelling of the overland flow losses. Other processes such as spray drift and volatilisation may also transport pesticides directly to surface water, but these were considered less important relative to the other pathways and therefore not included in the modelling.


The overland flow pesticide flux $L_{of}$ [M L$^{-2}$ T$^{-1}$] was calculated using Eq. 2 and assuming that the fraction of applied pesticide that will run off a given field and reach the reservoir ($AF_{of}$) can be estimated as:

$$AF_{of} = \exp\left(-\frac{t_{ro} * \ln(2)}{DT50}\right) * f_{slope} * f_{dist} * f_{buffer} \qquad (12)$$


where $t_{ro}$ [T] is the time passing between pesticide application and the occurrence of a runoff event, and $f_{slope}$, $f_{dist}$ and $f_{buffer}$ are the slope, distance and buffer correction factors, respectively, given by:

$$f_{slope} = \begin{cases} \dfrac{S}{10} & \text{, if } S < 10 \text{ degrees} \\ 1 & \text{, if } S \geq 10 \text{ degrees} \end{cases} \qquad (13)$$

$$f_{dist} = \begin{cases} \dfrac{1}{d_{fr}} & \text{, if } d_{fr} \geq 1 \text{ m} \\ 1 & \text{, if } d_{fr} < 1 \text{ m} \end{cases} \qquad (14)$$

$$f_{buffer} = 1 - E_{buffer} \qquad (15)$$

where S [degrees] is the local slope, $d_{fr}$ [L] is the distance from the field to the reservoir, and E [%] is a retention efficiency of a field buffer strip. The above approach for modelling overland flow attenuation follows on from REXTOX (OECD, 2000). In REXTOX, the time between pesticide application and the occurrence of a runoff event is assumed to be three days, whereas we assumed the time to depend on the month of application (see Appendix A). For simplicity and because we considered the contribution from both runoff and erosion, the available amount of pesticide available for run-off was not corrected for sorption
or for plant interception. The slope correction was assumed to be a linear function up until a local slope of 10 degrees beyond which no correction took place. The buffer correction factor was informed based on a review of buffer retention efficiencies (Reichenberger et al., 2007). It should be noted that REXTOX only considered pesticide losses via runoff from fields adjacent



to surface waters and therefore did not include the effect of distance from field to water body. Here, we assumed that the attenuation was inversely proportional to the distance from the reservoir.


The overland flow flux was evaluated similarly to the leaching risk (Eq. 11) to allow a comparison of the relative contribution of the two components to the combined risk.

The above presented approach enabled us to a) asses relative pesticide loss risk from all fields in the study catchment, b) compare overland flow risk to groundwater leaching risk for all fields and c) evaluate optimal spatial targeting and the effect of available mitigation measures in the whole study catchment.

### 2.5.3 Model implementation and testing

The model was constructed in GeNIe 3.0 (www.bayesfusion.com). Prior probabilities for network variables were calculated from data described in section 2.2 and Appendix A. Discrete variables were assigned a number of mutually exclusive 'states' with conditional probabilities captured in Conditional Probability Tables (CPTs). Prior probability distributions for continuous nodes were fitted to available data using the $5^{th}$, $50^{th}$ and $95^{th}$ percentiles of the cumulative probability distribution (O'Hagan, 2012) in the SHELF package (Oakley, 2020) in the open source statistical modelling software R (The R Project for Statistical Computing 4.0.1). A discretised version of the model was then exported to R and applied at field level, using the package bnspatial (Masante, 2017). A discretisation method selected for each node is described in Appendix A. Discretisation was based on a mix of expert opinion (e.g., soil organic carbon and hydraulic conductivity, as well as groundwater pesticide flux and the final surface water risk, which were discretised considering the likelihood of exceeding the drinking water standard concentration of 0.1 µg $L^{-1}$), accepted values in literature (e.g., pesticide $K_{oc}$ and half-life), uniform cases (e.g., rainfall, temperature, and PET), and uniform interval width (e.g. depth to groundwater, distance, slope). Child nodes such as AET, infiltration rate and overland flow attenuation were discretised using interpolation to ensure that conditional probabilities for the combination of parent node states (low/low, medium/medium, high/high) were meaningful. Application rate discretisation was based on equal counts but adjusted to ensure that change in application rates would result in a shift between risk classes, with the number of states maximised to allow sensitivity to change, while considering model run time.

Available spatial GIS layers were used as 'hard' evidence to set states for relevant nodes and produce spatially explicit simulations of probabilistic outcomes.

Uncertainty in the simulated outcomes in the spatial implementation of the model was evaluated by calculating the Shannon entropy index of the target nodes. The entropy H(X) for node X is defined as:




$$H(X) = -\sum P(X) \log_2 (P(X))$$

The entropy quantifies the information content within a node and equals 0 if X is known with certainty and is maximised when $X$ is unknown (i.e. $X$ is given by a uniform distribution).


Sensitivity analysis of the discretised model was undertaken in GeNIe using the algorithm of Kjærulff and van der Gaag (2000) that calculates a complete set of derivatives of the posterior probability distributions over the target nodes over each of the numerical parameters of the Bayesian network, using the two modelled risk pathways and the combined risk as target nodes. Euclidean distance measure, which quantifies the distance between the various conditional probability distributions over the

child node, conditional on the states of the parent node, was used to calculate the strength of influence between variables (Koiter, 2006). Simulated surface water risk (10,000 simulated values of surface water pesticide concentration in µg L[-1]) were compared with the limited available water quality observations for four active ingredients from month January – March (see Section 3.4). Model credibility was furthermore evaluated using stakeholder feedback.

**2.6 Simulated scenarios**

A questionnaire was used to elicit stakeholder feedback regarding potential alterations to the management of crops/pesticides and mitigation strategies from steering group members representing a grower, a regulator and a drinking water supplier in the study catchment to develop plausible pesticide mitigation scenarios. The agreed scenarios included:

- Baseline risk for five active ingredients
- Delayed pesticide application by one month (January to February and February to March)
- Reduced application rate by 10%, 25% and 50%
- Additional buffering of fields to reduce overland pesticide runoff
- Presence/absence of soil pan

**3.    Results and Discussion**

A number of detailed mechanistic models have successfully simulated pesticide dynamics at plot and catchment scale (Piffady et al., 2020). However, detailed observational data required for the calibration and validation of detailed models is not widely available to managers in many drinking water catchments. In this case study, in addition to sparse water quality observations, process-based modelling was hindered by the complex water transfers and limited gauging at the reservoir outlet, which

prevented the calculation of the hydrological balance due to lack of data on water transfers from neighbouring catchments and the productivity of the desalination plant. Difficulty with closing a water balance without considerable uncertainty is a known problem in many catchments that affects practical application of many modelling approaches (Beven et al., 2019). Therefore, to support decision making, we developed a probabilistic model of intrinsic vulnerability to pesticide pollution within a





Bayesian framework to allow the assessment of intrinsic pesticide risk and inform management. As pesticide risk assessment
is inherently uncertain, due to many complex and poorly characterised processes, the graphical BBN model helps to improve
the transparency of the risk management process (Carriger and Newman, 2012). Furthermore, the probabilistic assessment
provided by the BBN methodology is more in line with the classical definitions of risk than the more commonly used single-
value risk quotients (Moe et al., 2021).The model represents key processes to capture combined uncertainties stemming both
from observational data and limited knowledge (Sahlin et al., 2021).


### 3.1 Can we characterise the spatial and temporal variability of pesticide pollution risk from groundwater leaching and overland flow using limited observational data?

The causal structure of the hybrid BBN model designed in GeNie is shown in Figure 3. The network consists of 45 nodes and
75 arcs. The results of the spatial simulation of the groundwater leaching pesticide flux, the overland flow pesticide flux, and
the overall surface water risk are shown in Figures 4-6. The spatial application of the discretised model for five active
ingredients has mostly shown a uniform low degree of pesticide leaching to groundwater across the 3.1 km$^2$ study catchment,
with the exception of prosulfocarb applied at the highest application rate (Fig. 4).  The largely low groundwater leaching risk
is not surprising, due to the very high groundwater attenuation rates resulting from the considered pesticides being neither
particularly mobile (all have relatively high K$_{oc}$ values) nor persistent (all have relatively short half-lives) (Table 2). Piffady
et al. (2020) also found that sub-surface vulnerability was the least discriminating spatial layer, as compared to other
hydrological pathways.

Conversely, the overland flow pesticide fluxes showed a distinct spatial variability, with most risky fields located on the
steepest fields closest to surface water bodies (Fig. 5). Figure 5 also suggests that more of the risky fields are located around
the west stream, which agrees with the fact that the observed pesticide concentration levels in the west stream are generally
greater than in the east stream (cf. Table 1). This can be explained by more fields being treated with pesticides in the western
part of the catchment (i.e. more fields where the dominant land use is potato) but also by more permeable hydrogeological
formations (i.e. blown sand) and soils located in the south east part of the catchment, and hence less runoff expected to be
generated in this area (Fig. 1b). As the overland pesticide flux is more closely related to the final surface water risk than is the
groundwater pesticide flux (Fig. 7, Table 3), the resulting risk maps for overland flux and surface water risk flux look similar
(Figs. 5-6).

Entropy calculation for overland flow pesticide fluxes (Fig. 5) and surface water risk (Fig. 6) suggests that in the baseline
scenario, the risk assessment status class is more certain for the fields closest to the streams, while for the maximum
intervention scenario, the uncertainty is more evenly distributed across the catchment. The assessment of groundwater leaching
pesticide fluxes is generally more certain (Fig. 4). Regardless of the absolute values, the relative difference in entropy between





different management scenarios and risk status classes is informative for informing management interventions and safeguarding managers from putting too much or too little confidence in the final risk assessment (Sahlin et al., 2021).

The application of continuous and hybrid networks in environmental risk assessment is rare (Kaikkonen et al., 2021). The integration of BBNs with GIS for spatial risk assessment has recently been increasing but is still limited (Carriger et al., 2021; Guo et al., 2020; Kaikkonen et al., 2021; Pagano et al., 2018). A major advantage of the BBN approach presented here over existing index methods for pesticide risk assessment is that the risk and the associated uncertainty can be determined and mapped, thereby allowing the confidence in the results to be directly assessed. The hybrid network allows more detailed

characterisation of multiple processes and their uncertainty, than a typical index-based GIS method. However, the need to discretise the network for spatial application currently presents a major methodological limitation, leading to loss of information, and would merit further research and development.

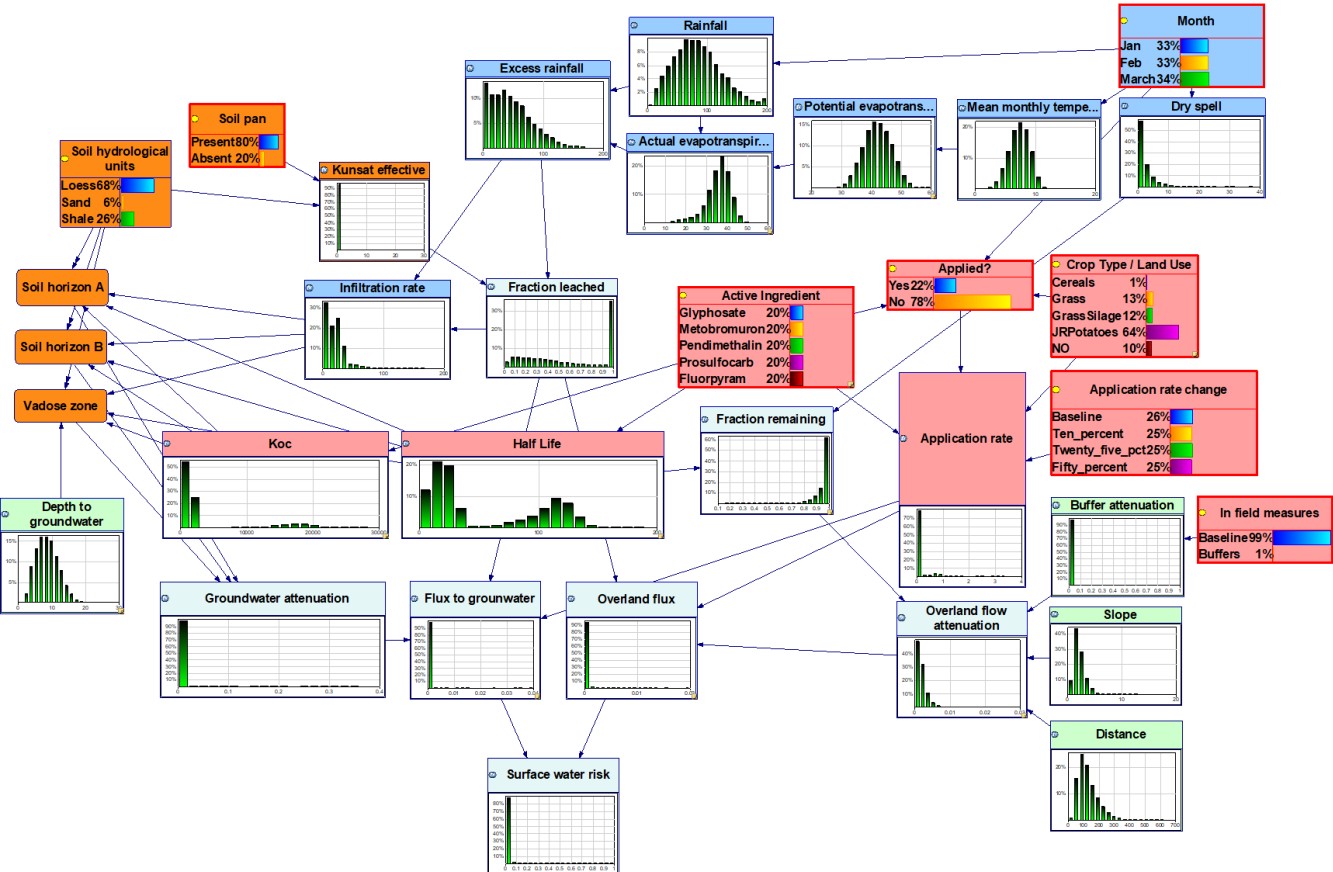

**Figure 3: Conceptual structure of the pesticide risk model. Blue = climate/hydrological variables; orange = soil variables; pale green = site-specific variables; red = land management/pesticide-specific variables. Nodes with thick red border show discrete variables that can be manipulated to model management scenarios. Histograms show continuous variables included in the model.**





**Figure 4:** Spatial variability and associated uncertainty (entropy) of groundwater leaching risk under current practices for all AI under baseline and maximum mitigation scenarios.







**Figure 5: Spatial variability and associated uncertainty (entropy) of overland runoff pesticide fluxes under current practices for the five active ingredients (Gl=glyphosate; Met=metobromuron; Pen=pendimethalin; Pr=prosulfocarb; Fl=fluopyram) under baseline and maximum mitigation scenarios.**




**Figure 6:** **Spatial variability and associated uncertainty (entropy) of surface water risk under current practices for the five active ingredients (Gl=glyphosate; Met=metobromuron; Pen=pendimethalin; Pr=prosulfocarb; Fl=fluopyram) under baseline and maximum mitigation scenarios.**





## 3.2 Which factors are most influential on intrinsic pesticide pollution risk?

Sensitivity analysis has identified the most influential parameters affecting the pesticide leaching and the overland flow pesticide fluxes. Figure 7 shows the result of sensitivity analysis graphically with nodes coloured in red being more important

for the calculation of the posterior probability of the pesticide risk nodes. Pesticide pollution risk was particularly sensitive to crop type, time of application, overland attenuation, slope and proximity to the surface water body. Crop type directly determines the expected amount of pesticide applied in a given field, whereas the time of year affects expected rainfall and length of a dry spell following application. Alongside the soil hydraulic conductivity, the latter two influential variables determine the amount of infiltration and overland flow. It is apparent that not all variables in the model contributed strongly

to the final risk assessment, hence model simplification may be possible. However, it may be advisable to test model transferability to other locations first to confirm these relationships, before omitting potentially uninfluential variables. For example, depth to groundwater appears to be uninfluential in this study catchment, which may be explained either by the relatively uniform shallow depths and uncertain hydrogeological data or by most pesticide retention and degradation taking place in the A and B soil horizons, with limited influence of the vadose zone. These hypotheses could be tested in a study

catchment with a better understanding of the sub-surface. Evapotranspiration calculations also appear to have limited impact and could potentially be omitted from the model for simplification.

Figure 7 shows the results of the strength of influence analysis, where the thickness of the arrows represents the strength of influence between two directly connected nodes based on the Euclidean distance between the probability distributions. The

top 20 most closely related variables with Euclidean distance > 0.5 are presented in Table 3. All the relationships are intuitive and build confidence in reliable specification of the conditional probabilities and hence in model simulations.





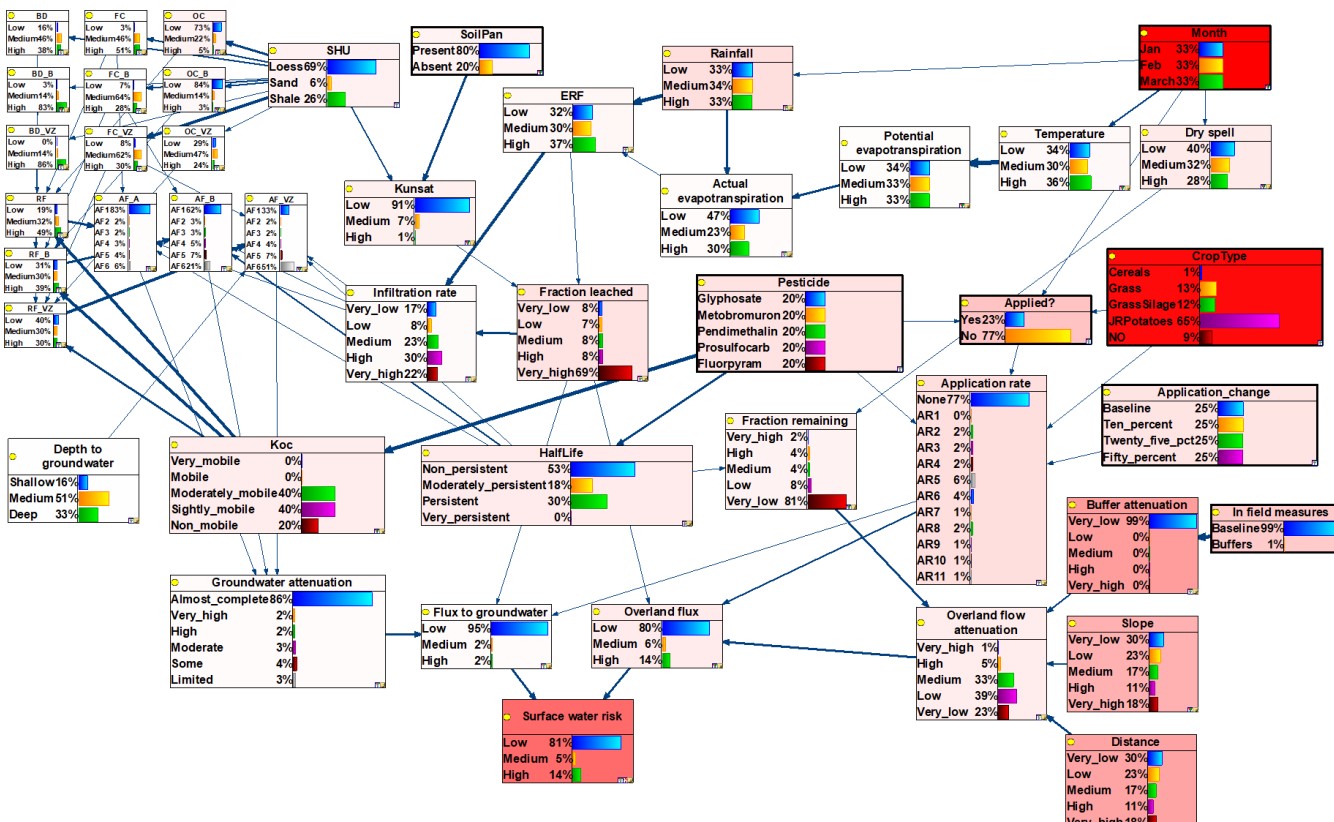

**Figure 7: Strength of Influence and sensitivity analysis (Kjærulff and van der Gaag, 2000) using surface water risk, Groundwater flux and Overland flow flux as target nodes. Deeper red colouring shows the more influential variables. Thickness of arrows indicate the strength of influence between two directly connected nodes calculated as Euclidean distance**





**Table 3. Strength of Influence for top twenty most closely related variables (Euclydean distance > 0.5). RF – retardation factor, AF – attenuation factor, VZ – vadose zone, B – soil horizon B.**

| Parent | Child | Average | Maximum | Weighted |
|---|---|---|---|---|
| Temperature | Potential evapotranspiration | 0.95 | 0.99 | 0.95 |
| Rainfall | Effective rainfall | 0.81 | 1.00 | 0.81 |
| Pesticide | Koc | 0.80 | 1.00 | 0.80 |
| In field measures | Buffer attenuation | 0.76 | 0.76 | 0.76 |
| $K_{oc}$ | RF | 0.67 | 1.00 | 0.67 |
| $K_{oc}$ | RF_B | 0.67 | 1.00 | 0.67 |
| Effective rainfall | Infiltration rate | 0.64 | 0.88 | 0.64 |
| Soil hydrological unit | FC_VZ | 0.63 | 0.87 | 0.63 |
| RF_VZ | AF_VZ | 0.61 | 1.00 | 0.61 |
| Pesticide | Half Life | 0.59 | 0.98 | 0.59 |
| Overland flow attenuation | Overland flux | 0.59 | 1.00 | 0.59 |
| $K_{oc}$ | RF_VZ | 0.58 | 1.00 | 0.58 |
| RF_B | AF_B | 0.57 | 0.99 | 0.57 |
| Potential evapotranspiration | Actual evapotranspiration | 0.56 | 0.91 | 0.56 |
| Soil hydrological unit | Organic carbon | 0.56 | 0.79 | 0.56 |
| Groundwater attenuation | Flux to groundwater | 0.53 | 1.00 | 0.53 |
| Flux to groundwater | Surface water risk | 0.52 | 0.96 | 0.52 |
| Fraction leached | Infiltration rate | 0.51 | 0.93 | 0.51 |
| SoilPan | $K_{unsat}$ | 0.51 | 0.96 | 0.51 |
| Overland flux | Surface water risk | 0.51 | 0.95 | 0.51 |


### 3.3 What is the effectiveness of available management interventions on pesticide risk reduction?

The BBN model was applied to evaluate the effectiveness of the following mitigation measures on reducing the pesticide risks: delayed timing of pesticide application; 10%, 25% and 50% reduction in application rate; additional field buffers; and presence/absence of soil pan. Figure 8-9 show the results of the simulated management scenarios on, respectively, the overland

flow pesticide flux of metobromuron and the groudwater leaching pesticide flux of prosulfocarb (similar results for the other pesticides can be found in the Appendix B).

The time between pesticide application and the first runoff event is often considered critical for the mobilisation and loss of pesticide via runoff, and hence, avoiding application in months with a higher probability of runoff events can potentially lead

to a reduction in risk. The figures suggest that delaying the application of pesticide until March results in a decrease in runoff risk for metobrouron but has limited impact on the leaching of prosulfocarb to groundwater, as groundwater risk is not related to the length of a dry spell and associated pesticide degradation following pesticide application.





Reduction in application rates unsurprisingly results in a reduced risk, particularly in the groundwater leaching risk (Figure ),
which is reduced to Low even after 10% reduction. Introduction of buffers reduces the runoff risk to a similar extent as a 50%
reduction of pesticide application rates for overland flow risk and hence may be a more cost-effective mitigation intervention.
Managing and removing potential plough pans increases the amount of infiltration into soils, thus reducing the pesticide runoff
risk to an extent that is comparable to 10% reduction in application rates (Fig. 8). By combining all available maximum
interventions of 50% reduction in pesticide application rate, management of plough pan, delayed application timing and
installing additional field buffers, the probability of all types of risk is notably reduced (Figs. 4-6, 8-9).

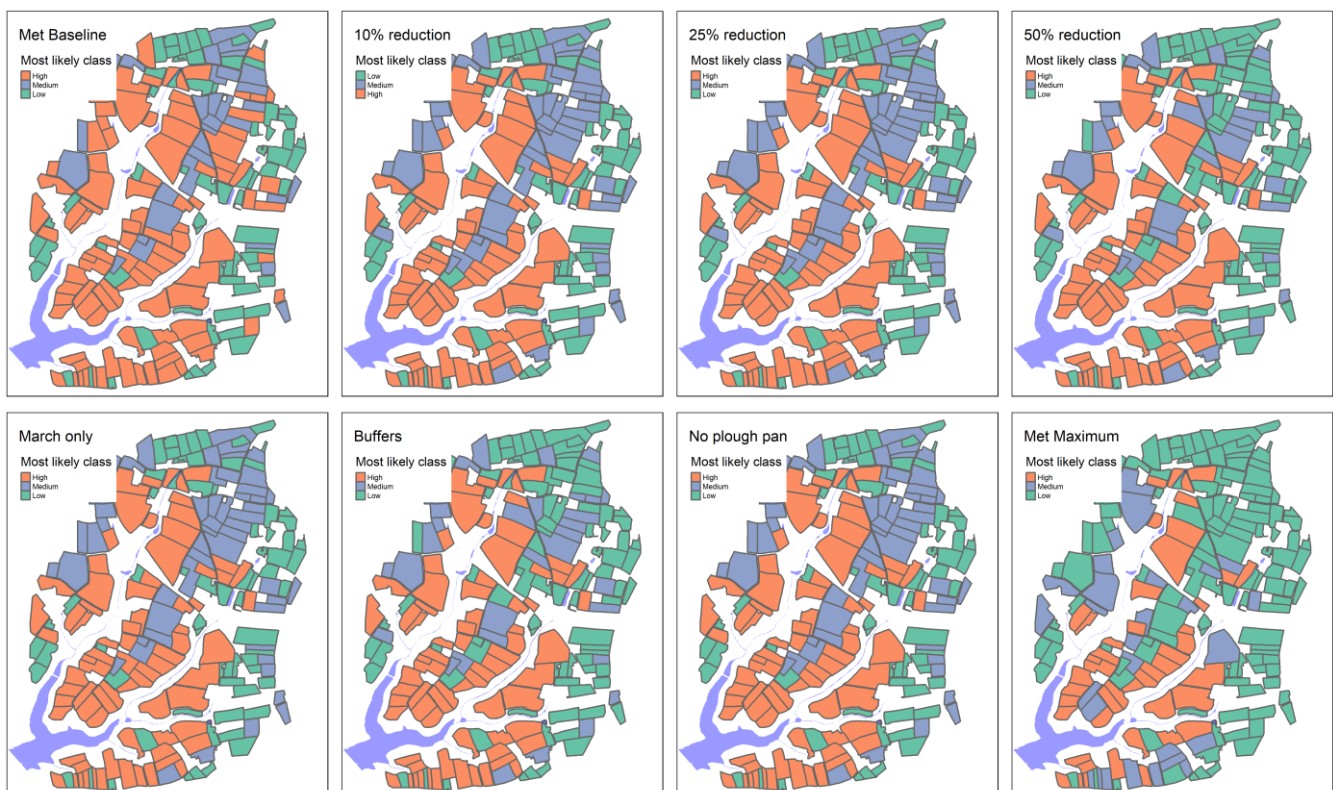

**Figure 8: Example output showing most likely overland flow risk class for each field for Metobromuron under current application**
**practices, 10% reduction, 25% reduction, 50% reduction, time shift of application to March, additional field buffers, no plough pan,**
**combined all available mitigation measures – shifting application to March, 50% reduction in application rates, buffers and no soil**
**pan**.



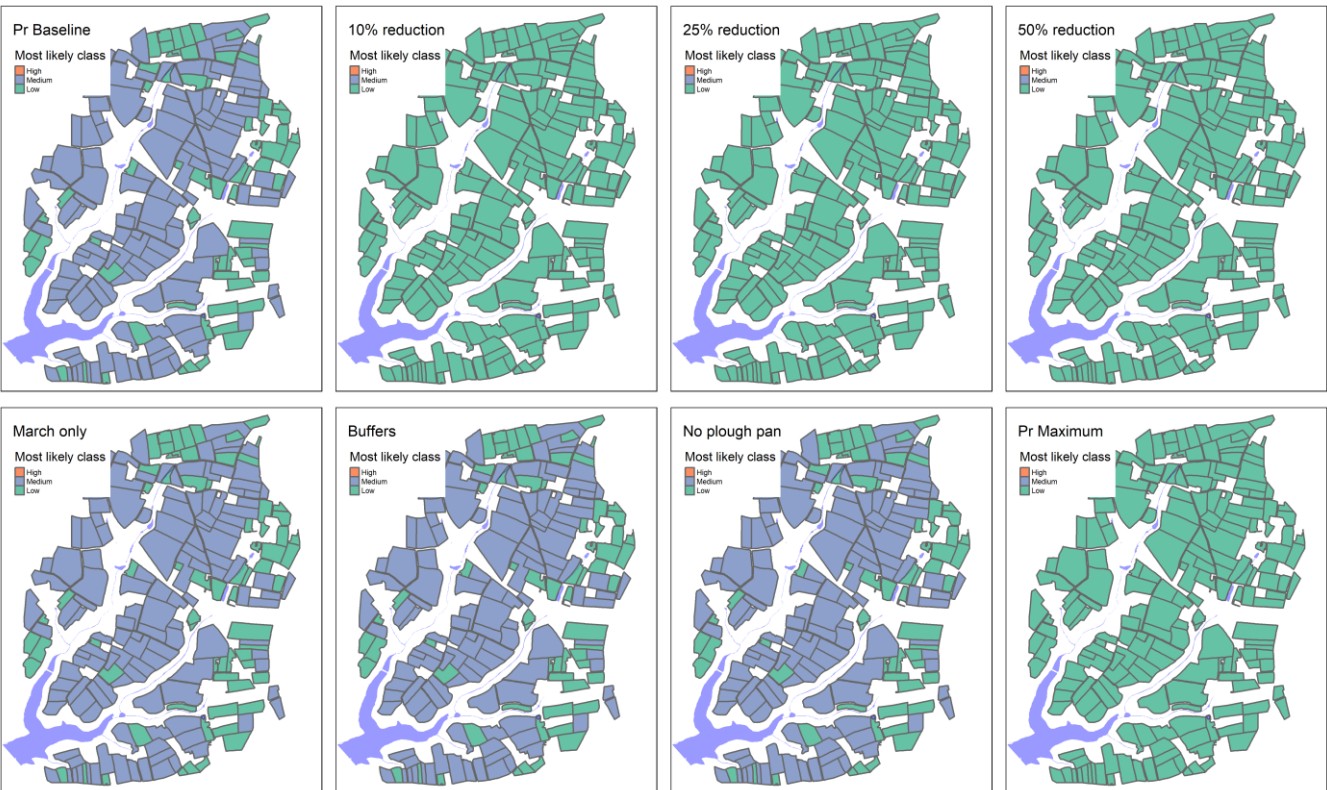

**Figure 9:** **Example output showing most likely groundwater leaching risk class for each field for Prosulfocarb under current application practices, 10% reduction, 25% reduction, 50% reduction, time shift of application to March, additional field buffers, no plough pan, combined all available mitigation measures – shifting application to March, 50% reduction in application rates, buffers and no soil pan.**

## 3.4 Model validation

Figure 10 shows a comparison between the probability density distributions based on 10,000 surface water risk simulations for each active ingredient and the limited observational data (in µg L⁻¹) available for the months January – March between 2016 and 2019. The model typically over-estimates the simulated risk for glyphosate and pendimethalin, albeit with low probability of high values. The simulated and observed distributions for prosulforcarb are comparable, whilst the model seems to under-estimate the risk from metobromuron. However, it has to be noted that the very few observations available for metobromuron,(N=8) seem to be higher and less accurate than for the other pesticides. It should also be noted that the developed model was never intended to represent the complex transport and fate processes in the catchment in detail or to accurately simulate the pesticide concentration levels in the reservoir, so the comparison in Figure 10 was mainly carried out as a sense check of the model predictions. Overall, model simulations appear conservative, which is helpful in terms of



informing a precautionary management approach. Further model refinement could focus on constraining the upper simulation
values throughout the model.

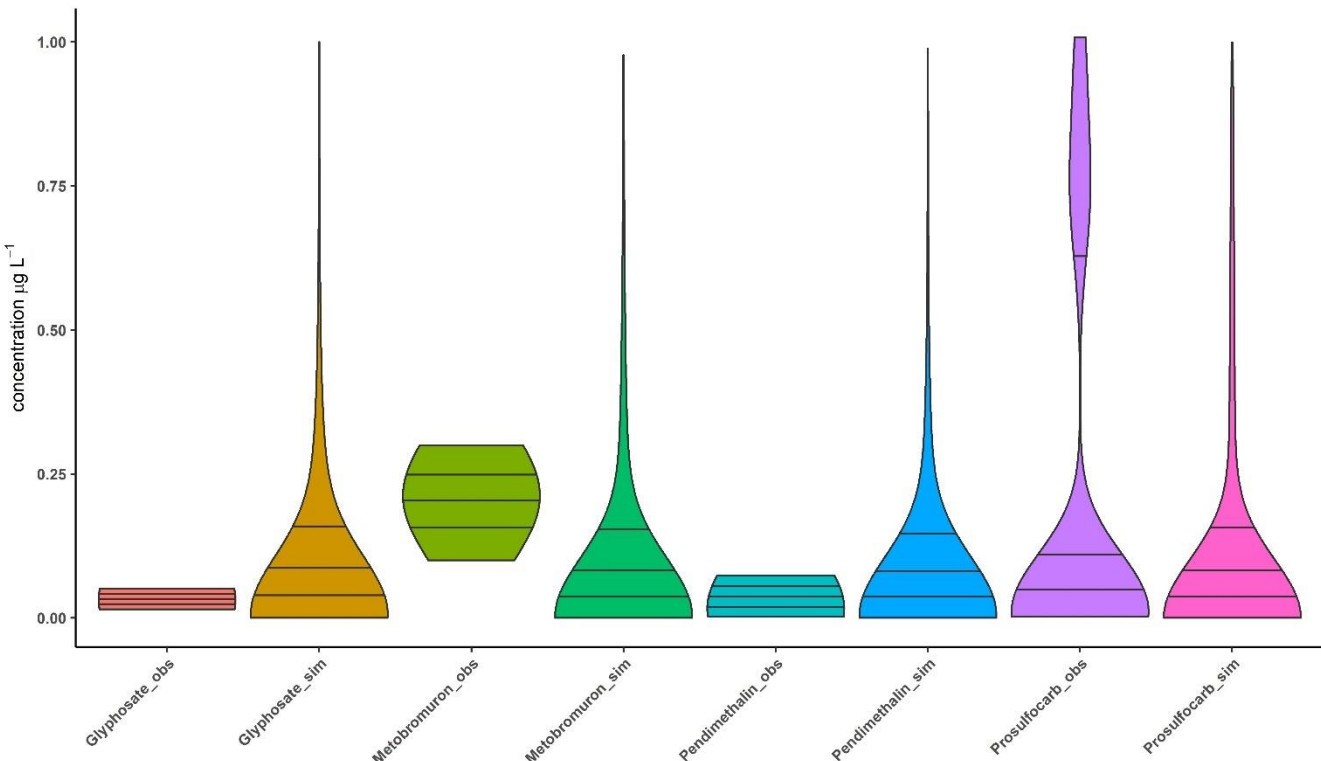

**Figure 10: Violin plots showing the probability density distribution and three quartiles (25th, 50th, 75th) of simulated (N=10,000 iterations) vs. observed (glyphosate N=20, metobromuron N=8, pendimethaling N=73 and prosulfocarb**
**N=25) concentrations for four active ingredients.**

### 3.5 Limitations and outlook

A unique advantage of a BBN is the ability to inform probabilistic decisions on the basis of incomplete data (Panidhapu et al., 2020) and address 'what-if' counterfactual scenarios (Gibert et al., 2018; Moe et al., 2021) as well as the ability to integrate
data of different quality from different sources and disciplines. In machine learning, the selected mathematical approach needs to be based on the target question to be answered and be aligned with the properties of the available data (Gibert et al., 2018). This study presents a novel approach to pesticide risk analysis that matches the question in hand with sparse data in a poorly monitored drinking water catchment. We constructed a causal network, where the model structure was informed by expert knowledge. However, Bayesian Networks can also be used as machine-learning associative tools that are suitable for deriving
patterns in datasets without a specific response variable. It could be argued that pesticide risk, expressed as flux or concentration of pesticide in different potential loss pathways (overland flow or groundwater leaching) is a latent variable



without available observational data (Piffady et al., 2020), making it difficult to calibrate or validate a risk model. Hence, model credibility and salience (Cash et al., 2005) need to be evaluated by experts and stakeholders. Here, we implemented a simple validation approach to confirm that the model predictions fall within the realms of credibility. Further validation
approaches could be explored in future implementations. The spatial application in the R package bnspatial allows to simulate expected quantities, based on the median value of each discretisation interval. Hence, by multiplying the combined fluxes from each field with the field areas and summing the resulting pesticide masses over all fields in the catchment and dividing by the reservoir volume, a concentration in the reservoir water could be estimated for each pesticide and month, which could be compared to measured concentrations if available for further model validation. However, this deterministic calculation would
be heavily reliant on the discretisation of the target node in question, and, coupled with rare extreme high values generated by the stochastic model, would make the validation uncertain. Hence, it would be best applied in combination with dynamic discretisation, if available, and with further model development constraining upper simulated values.

The BBN model could easily be extended to consider other pesticides by changing the pesticide-specific properties, while
greater structural changes would be needed to simulate the cumulative risk from total pesticide concentrations. However, the developed BBN also has several limitations and there are various input parameters and elements that could be refined and improved. The BBN focuses only on aquatic risks resulting from intentional application of pesticide in agriculture and does not consider potential point sources of pesticide contamination (such as misuse, accidental spillages, disposal of pesticides or cleaning of application equipment). Although quite detailed in process representation that is based on established mechanistic
approaches, the modelling of pesticide leaching and runoff in the BBN is still simplified and could be extended to consider e.g. preferential flow pathways. Soil water balance and hydrology are critical for pesticide risk assessments, the details of which are challenging to capture with a BBN or an index-based model. Hence, future developments of the approach could include development of an improved probabilistic soil hydrology model linked to the pesticide risk model.

Discretisation is recognised as a major limitation of BBNs (Nojavan A. et al., 2017). Whilst here we constructed a hybrid network that has largely allowed us to avoid the loss of information associated with discretisation, this advantage was lost in the spatial application where existing mathematical and software limitations prevent direct coupling of a hybrid network with GIS. We suggest that this limitation could be an interesting and fruitful avenue for further research and methodological development, e.g. by developing software applications that allow automated dynamic discretisation (Fenton and Neil, 2013)
coupled with GIS. Finally, model validation would be helped by confronting with field data in a highly monitored experimental catchment, thus also allowing to evaluate the model transferability.

Notwithstanding the above limitations, this modelling approach satisfies many of the requirements of an 'ideal' model to support environmental decision making, as set out by Schuwirth et al. (2019), in that it 'can be directly linked to management
objectives, predicts effects of management alternatives without bias, includes adequate precision and a correct estimate of



prediction uncertainty .. and is easy to understand'. We also developed the model with the final criterion of 'easy transferability in space and time' in mind, and this could be tested in future applications.

## 4.    Conclusions

In this study we present a spatial Bayesian Belief Network (BBN) that simulates inherent pesticide risk to groundwater and surface water quality, identifies critical source areas and informs field-level pesticide mitigation strategies in a small drinking water catchment with limited observational data. The BBN accounted for the spatial heterogeneity of surface water risk from pesticides, taking into account the spatial distribution of   soil properties (texture, organic matter content, hydrological properties), topographic connectivity (slope, distance to surface water/depth to groundwater) and agronomic practices;

temporal variability of climatic and hydrological processes (temperature, rainfall, evapotranspiration, overland and subsurface flow) as well as uncertainties related to pesticide properties and the effectiveness of management interventions. The risk of pesticide loss via overland flow and leaching to groundwater were simulated for five active ingredients. Overland pesticide pollution risk from overland flow showed clear spatial variability across the study catchment, while groundwater leaching risk was more uniform. The effectiveness of mitigation measures such as delayed timing of pesticide application, reduction in

application rates, installation of additional field buffers; and management of soil plough pan on risk reduction were evaluated. Combined interventions of 50% reduced pesticide application rate, management of plough pan, delayed application timing and field buffer installation greatly reduced the probability of high-risk from overland flow. The advantages of the presented BBN approach over traditional index-based methods include its ability to integrate diverse data sources (both qualitative and quantitative) for a field-scale assessment of critical source areas of pesticide pollution in a data sparce catchment, with explicit

representation of uncertainties. The graphical nature of the decision support tool facilitates interactive model development and evaluation with stakeholders to build model credibility; while its flexible and dynamic nature allows for performing both predictive and diagnostic reasoning based on observations, which can be linked to spatially explicit data, thus improving pesticide risk management.

## Appendix A Model description

**Table A1: Definition of model variables included in the Bayesian Belief Network. Definition of states and boundaries as well as the information and assumptions used to populate prior probabilities or conditional probability tables for each node.**

| Variable (symbol) [unit] | States | Boundaries | Description |
|---|---|---|---|
| **Soil and site-specific variables** | | | |
| Soil hydrological units (SHU) | Loess | | Soil hydrological units observed in the Val de la Mar catchment. The proportion of SHUs in the catchment is derived from the hydrogeological map of Jersey (Robins et al. 1991): Loess (68.7%), Sand (5.7%) and Shale (25.6%) |
| | Sand | | |
| | Shale | | |
| | Low | <1.2 | |





| Bulk density topsoil (BD) [g cm⁻³] | Medium | 1.2 - 1.4 | Bulk density of topsoil, subsoil and parent material [g cm$^{-3}$], respectively. The bulk densities for each soil hydrological unit and horizon have been derived from HYPRES database by fitting a truncated normal distribution with mean ($\mu$) and standard deviation ($\sigma$) to the data, truncating the lower tail at the minimum value in the HYPRES database. Discretisation boundaries were based on expert opinion. |
|---|---|---|---|
| | High | >1.4 | |
| Bulk density subsoil (BD_B) [g cm⁻³] | Low | < 1.2 | |
| | Medium | 1.2 - 1.4 | |
| | High | >1.4 | |
| Bulk density parent material/vadose zone (BD_VZ) [g cm⁻³] | Low | <1.2 | |
| | Medium | 1.2 - 1.4 | |
| | High | >1.4 | |

| Bulk density | Loess | Sand | Shale |
|---|---|---|---|
| Topsoil | μ=1.40; σ=0.11 | μ=1.30; σ=0.10 | μ=1.20; σ=0.17 |
| Subsoil | μ=1.53; σ=0.10 | μ=1.55; σ=0.14 | μ=1.47; σ=0.24 |
| Vadose zone | μ=1.48; σ=0.08 | μ=1.64; σ=0.08 | μ=1.52; σ=0.19 |

| Field capacity topsoil (FC) | Low | < 0.3 | Field capacity of topsoil, subsoil and parent material, respectively. The field capacity for each soil hydrological unit and horizon have been derived from the HYPRES database and assuming field capacity is the water content at -50 cm pressure head. A truncated normal distribution with mean ($\mu$) and standard deviation ($\sigma$) have been fitted to the data, truncating the lower tail at the minimum value in the HYPRES database. Discretisation boundaries were based on expert opinion. |
|---|---|---|---|
| | Medium | 0.3 - 0.4 | |
| | High | >0.4 | |
| Field capacity subsoil (FC_B) | Low | < 0.3 | |
| | Medium | 0.3 - 0.4 | |
| | High | >0.4 | |
| Field capacity parent material/vadose zone (FC_VZ) | Low | < 0.3 | |
| | Medium | 0.3 - 0.4 | |
| | High | >0.4 | |

| Field capacity | Loess | Sand | Shale |
|---|---|---|---|
| Topsoil | μ=0.40; σ=0.03 | μ=0.31; σ=0.08 | μ=0.42; σ=0.07 |
| Subsoil | μ=0.37; σ=0.05 | μ=0.28; σ=0.05 | μ=0.35; σ=0.06 |
| Vadose zone | μ=0.39; σ=0.03 | μ=0.23; σ=0.02 | μ=0.35; σ=0.05 |

| Organic carbon topsoil (OC) [%] | Low | <2 | Organic carbon content of topsoil, subsoil and parent material [%]. The organic carbon content for each soil hydrological unit and horizon is derived from organic matter content (OM) data in the HYPRES database and assuming that the organic carbon fraction of organic matter is 58%, i.e.: $OC = OM/1.724$ The organic carbon content is assumed to follow a normal distribution with mean ($\mu$) and standard deviation ($\sigma$), with the normal distributions being truncated at the respective minimum values in the HYPRES database. Discretisation boundaries were based on expert opinion. |
|---|---|---|---|
| | Medium | 2-4 | |
| | High | >4 | |
| Organic carbon subsoil (OC_B) [%] | Low | < 1 | |
| | Medium | 1-2 | |
| | High | >2 | |
| Organic carbon vadose zone (OC_VZ) [%] | Low | <0.2 | |
| | Medium | 0.2-0.5 | |
| | High | > 0.5 | |

| Organic carbon | Loess | Sand | Shale |
|---|---|---|---|
| Topsoil | μ=1.04; σ=0.25 | μ=3.02; σ=1.33 | μ=2.78; σ=1.04 |
| Subsoil | μ=0.24; σ=0.17 | μ=0.77; σ=0.60 | μ=0.86; σ=0.79 |
| Vadose zone | μ=0.23; σ=0.25 | μ=0.21; σ=0.26 | μ=0.38; σ=0.35 |

| Soil pan (SoilPan) | present | | The presence of a low-permeable soil pan is believed to be widespread in the catchment. The prior distribution is assumed as: present (80%), absent (20%), based on stakeholder feedback. |
|---|---|---|---|
| | absent | | |
| Effective unsaturated hydraulic conductivity (Kunsat) | Low | < 0.5 | The effective unsaturated hydraulic conductivity of the soil horizons (K$_{unsat}$) is calculated as the harmonic average of the conductivity of each horizon: |
| | Medium | 0.5 - 5 | |
| | High | > 5 | |



| [cm day⁻¹] | | | $K_{unsat} = \dfrac{d}{\sum_{i=1}^{N} d_i/K_{unsat\_i}}$ <br><br> where d is the total soil profile depth of the topsoil and subsoil combined (assumed to be 60 cm), $d_i$ and $K_{unsat\_i}$ are the thickness and unsaturated hydraulic conductivity of horizon i, respectively. The calculation can take presence of a soil pan into account. Soil hydrological unit and horizon specific unsaturated hydraulic conductivities are derived from the HYPRES database and depends on whether a soil pan is present (SP) or not (No SP). The hydraulic conductivity is assumed to follow a log-normal distribution with mean (μ) and standard deviation (σ) based on values in the HYPRES database. Discretisation boundaries were based on expert opinion. <br><br> Values of $\ln(K_{unsat\_i})$ [cm day⁻¹] is given below. SP=soil pan. |
|---|---|---|---|

|  |  | | Topsoil | Subsoil | Soil pan |
|---|---|---|---|---|---|
| Thickness [cm] | | No SP | 30 | 30 | 0 |
| | | SP | 30 | 20 | 10 |
| ln(K_unsat_i) [cm day⁻¹] | Loess | No SP | μ=-0.4; σ=1.7 | μ=-0.2; σ=1.1 | |
| | | SP | μ=-1.4; σ=1.7 | μ=-1.2; σ=0.9 | μ=-3.9; σ=0 |
| | Sand | No SP | μ=2.3; σ=0.4 | μ=3.2; σ=1.3 | |
| | | SP | μ=-0.9; σ=0.6 | μ=0.8; σ=1.2 | μ=-3.9; σ=0 |
| | Shale | No SP | μ=-1.9; σ=0.7 | μ=-1.1; σ=0.5 | |
| | | SP | μ=-4.0; σ=0.7 | μ=-2.9; σ=1.3 | μ=-3.9; σ=0 |

| Depth to groundwater (Depth) [m] | Shallow | ≤ 5 | Depth to groundwater is derived from the hydrogeological contours for the VDLM catchment provided by Jersey Water based on British Geological Survey data. A normal distribution has been fitted to this information (μ=7.9; σ=3.9) with the lower tail truncated at 2.5 m. Discretisation boundaries were based on equal intervals and expert opinion. |
|---|---|---|---|
| | Medium | 5 - 10 | |
| | Deep | >10 | |
| Distance [m] | Very low | < 50 | Distance to reservoir is derived by calculating the horizontal distance to the stream features using the Distance to nearest hub tool in QGIS. The distance was calculated from the polygon edge (i.e. vertex) that was nearest to the stream feature. A lognormal distribution was fitted to this information (μ=4.75; σ=0.54). Discretisation boundaries were based on equal intervals and expert opinion. |
| | Low | 50 -100 | |
| | Medium | 100 - 150 | |
| | High | 150 - 200 | |
| | Very high | >200 | |
| Slope | Very low | < 1.5 | Slope [degrees] is derived from a hydrologically-corrected digital terrain model (DTM) of 1m grid resolution. A log-normal distribution has been fitted to this information (μ=0.65; σ=0.49). Discretisation boundaries were based on equal intervals and expert opinion. |
| | Low | 1.5 - 2 | |
| | Medium | 2 - 2.5 | |
| | High | 2.5 - 3 | |
| | Very high | >3 | |
| **Climatic and hydrological variables** | | | |
| Month | Jan, Feb, Mar | | |
| Rainfall (Rainfall) | Low | < 65 | |

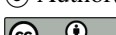


| [mm month⁻¹] | Medium | 65 - 100 | Monthly rainfall is derived from observed rainfall data (1894-2019) from the Government of Jersey website (https://opendata.gov.je/). Discretisation was based on uniform counts. |
| | High | >100 | |
| Mean monthly temperature (Temperature) [C] | Low | < 6.6 | Mean monthly temperature is derived from observed temperature data (1894-2019) from the Government of Jersey website (https://opendata.gov.je/). Discretisation was based on uniform counts. |
| | Medium | 6.6 - 8 | |
| | High | >8 | |
| Dry spell [days] | Low | ≤ 1 | The time between pesticide application and the first runoff event is determined by the likely length of a dry spell. Probability of dry spell length is calculated from daily rainfall data from 2014-2019 using the method by Hills and Morgan (1981) and assuming days with less than 0.25 mm rainfall are dry. |
| | Medium | 1 - 3 | |
| | High | >3 | |
| Potential evapotranspiration (PET) [mm month⁻¹] | Low | <40 | Potential evapotranspiration is calculated based on the Langbein formula (Pistocchi et al. 2006) : $PET = (300 + 25 * T + 0.05 * T^3)/12$ Equation is valid for calculating annual average PET, but it is here assumed applicable for calculating monthly PET based on monthly average temperature T. Discretisation was based on uniform counts. |
| | Medium | 40 - 45 | |
| | High | >45 | |
| Actual evapotranspiration (AET) [mm month⁻¹] | Low | <35 | Actual evapotranspiration is calculated from PET and rainfall based on the Turc method (Pistocchi, A., Pilar, V., Pennington, 2006). It is assumed this equation is valid for calculating monthly AET based on monthly rainfall and PET. $$AET = Rainfall * \left(0.9 + \left(\frac{Rainfall}{PET}\right)^2\right)^{-0.5}$$ Discretisation was based on uniform counts and interpolation to ensure that conditional probabilities for combination of parent node states are meaningful. |
| | Medium | 30-40 | |
| | High | >40 | |
| Excess rainfall (ERF) [mm month⁻¹] | Low | <30 | Monthly excess rainfall is the difference between rainfall and actual evapotranspiration and is calculated as follows: $$ERF = \begin{cases} 0, & \text{for Rainfall<AET} \\ Rainfall - AET, & \text{for Rainfall1AET} \end{cases}$$ Discretisation was based on uniform counts and interpolation to ensure that conditional probabilities for combination of parent node states (low/low. medium/medium, high/high) are meaningful. |
| | Medium | 30 – 60 | |
| | High | >60 | |
| Fraction leached ($f_{leach}$) | Very low | < 0.2 | Fraction of excess rainfall that will infiltrate to groundwater. It is assumed that all excess rainfall is infiltrating up to a maximum rate equal to $K_{unsat}$ $$f_{leach} = \begin{cases} 1, & \text{for ERF<}K_{unsat} * 30 * 10 \\ K_{unsat} * 10 * 30/ERF, & \text{for ERF≥}K_{unsat} * 30 * 10 \end{cases}$$ Discretisation was based on equal intervals. |
| | Low | 0.2 - 0.4 | |
| | Medium | 0.4 – 0.6 | |
| | High | 0.6 – 0.8 | |
| | Very high | >0.8 | |
| Infiltration rate (IR) [mm month⁻¹] | Very low | < 10 | The infiltration rate to groundwater $$IR = EFR * f_{leach}$$ Discretisation was based on interpolation to ensure that conditional probabilities for combination of parent node states are meaningful. |
| | Low | 10 - 15 | |
| | Medium | 15 - 30 | |
| | High | 30 - 60 | |
| | Very high | >60 | |
| **Pesticide, land use and land management variables** | | | |
| Crop Type/ Land Use | Cereals | | Land use and crop types are based on agronomic data provided by Jersey Royal as described in Section 3.3. |
| | Grass | | |
| | Grass/ Silage | | |





| | | | |
|---|---|---|---|
| | JRPotatoes | | |
| | NO | | |
| Active Ingredient | Glyphosate | | |
| | Metobromuron | | |
| | Pendimethalin | | |
| | Prosulfocarb | | |
| | Fluorpyram | | |
| $K_{oc}$ [L/kg] | Very mobile | 0 - 15 | $K_{oc}$ values and state classes are based on the University of Hertfordshire database. See Table 2 for typical application rates and properties of the selected pesticides. To account for uncertainty, the $K_{oc}$ values for each pesticide have been assumed to follow a normal distribution as stated below. Discretisation boundaries were based on accepted values (Lewis et al., 2016). |
| | Mobile | 15 - 75 | |
| | Moderately mobile | 75 - 500 | |
| | Sightly mobile | 500 - 4000 | |
| | Non-mobile | >4000 | |

| Active ingredient | $K_{oc}$ |
|---|---|
| Glyphosate | $\mu=1420$; $\sigma=232$ |
| Methobromuron | $\mu=197$; $\sigma=0.86$ |
| Pendimethalin | $\mu=17500$; $\sigma=3120$ |
| Prosulfocarb | $\mu=1690$; $\sigma=140$ |
| Flyopyram | $\mu=279$; $\sigma=19.8$ |

| | | | |
|---|---|---|---|
| Half Life [days] | Non-persistent | < 30 | Half-life values and state classes are based on the University of Hertfordshire database. See Table 2 for typical application rates and properties of the selected pesticides. To account for uncertainty, the half-life values for each pesticide have been assumed to follow a normal distribution as stated below. Discretisation boundaries were based on accepted values (Lewis et al., 2016). |
| | Moderately persistent | 30 - 100 | |
| | Persistent | 100 - 365 | |
| | Very persistent | >365 | |

| Active ingredient | Half-life |
|---|---|
| Glyphosate | $\mu=23.8$; $\sigma=7.52$ |
| Methobromuron | $\mu=22.4$; $\sigma=7.31$ |
| Pendimethalin | $\mu=101$; $\sigma=26.5$ |
| Prosulfocarb | $\mu=9.8$; $\sigma=1.39$ |
| Flyopyram | $\mu=119$; $\sigma=11$ |

| | | | |
|---|---|---|---|
| Application | Applied | | Intermediate variable that determines if a given pesticide is applied at a given month and for a given crop type. This has been populated based on the details provided in the separate section in the report on typical application rates and properties of the selected pesticides. |
| | Not applied | | |
| Application rate (AR) [kg ha$^{-1}$ yr$^{-1}$] | None | 0 – 1E-7 | Pesticide-specific application rates. This has been populated based on the details provided in the separate section 2.2 on typical application rates and properties of the selected pesticides. The application rates are adjusted depending on |
| | AR1 | 1E-7 - 0.1 | |
| | AR2 | 0.1 - 0.2 | |





| | AR3 | 0.2 – 0.3 | 'Application rate change' node. Discretisation was based on equal counts but adjusted to ensure that change in application rates would result in a shift between risk classes. The number of states was maximised to allow sensitivity to change while considering model run time. |
|---|---|---|---|
| | AR4 | 0.3 – 0.5 | |
| | AR5 | 0.5 – 0.8 | |
| | AR6 | 0.8 – 1.0 | |
| | AR7 | 1.0 – 1.2 | |
| | AR8 | 1.2 – 2.0 | |
| | AR9 | 2.0 – 2.5 | |
| | AR10 | 2.5 – 3.0 | |
| | AR11 | 3.0 – 4.0 | |
| Application rate change [%] | Baseline (0%) | | Management node |
| | 10% | | |
| | 25% | | |
| | 50% | | |
| In field measures (Measures) | Baseline | 0.99 | Management node that allows to simulate the effect of additional buffer implementation to reduce overland pesticide runoff from fields. |
| | Buffers | 0.01 | |
| Buffer attenuation ($E_{buffer}$) | Very low | <0.2 | Proportion of pesticide delivered to water course, conditioned on Measures. Modelled as a beta distribution on 0-1 scale, based on Reichenberger et al. (2007). Zero under baseline conditions and Beta(1.77,0.869 with additional buffers. Discretisation was based on equal intervals. |
| | Low | 0.2-0.4 | |
| | Medium | 0.4-0.6 | |
| | High | 0.6-0.8 | |
| | Very high | >0.8 | |
| **Calculated variables** | | | |
| RF A [unitless] | Low | <10 | Retardation factor for the topsoil, subsoil and parent material/vadose zone. The retardation factor describes the velocity of the solute pesticide relative to the infiltrating water. Hence, a RF=1 corresponds to a solute not experiencing any retardation due to adsorption (e.g. a tracer), whereas a RF=4 means that the solute travels 4 times slower than the infiltrating water etc. For a given pesticide and soil horizon, the retardation factor is calculated as:$$RF = 1 + \frac{BD * OC * K_{oc}}{FC}$$Discretisation boundaries were based on expert opinion. |
| | Medium | 10 - 50 | |
| | High | >50 | |
| RF B | Low | <10 | |
| | Medium | 10 - 50 | |
| | High | >50 | |
| RF VZ | Low | <10 | |
| | Medium | 10 - 50 | |
| | High | >50 | |
| AF A [unitless] | AF1 | 0 - 1E-5 | The attenuation factor during vertical solute transport through the topsoil (AF A). The calculation assumes 1D plug flow transport with the infiltrating water, linear retardation and first-order decay (Stenemo et al., 2007):$$AF\_A = \exp\left(\frac{-\ln(2) * RF\_A * dA * FC\_A}{\frac{IR}{30} * HalfLife} * 1000\right)$$where dA is the thickness of the topsoil, assumed to be 0.3 m. AF_A is the fraction of the applied pesticide that will reach the bottom of the topsoil and can take values between 0 (none of the applied pesticide will reach the bottom of the horizon) and 1 (all the applied pesticide will pass through the horizon). Discretisation boundaries were based on expert opinion on a logarithmic scale to reflect the skewed distribution. |
| | AF2 | 1E-5 – 0.0001 | |
| | AF3 | 0.0001 - 0.0.001 | |
| | AF4 | 0.001 – 0.01 | |
| | AF5 | 0.01 – 0.1 | |
| | AF6 | 0.1 - 1 | |





| AF B [unitless] | AF1 | 0 - 1E-5 | Attenuation factor during vertical solute transport through the subsoil (see details above). $$AF\_B = \exp\left(\frac{-\ln(2) * RF\_B * dB * FC\_B}{\frac{IR}{30} * HalfLife * 4} * 1000\right)$$ where dB is the thickness of the subsoil, assumed to be 0.3 m. It is assumed that the half-life during transport through the B horizon is 4 times longer than in the topsoil. Discretisation boundaries were based on expert opinion; a logarithmic scale was used to reflect the skewed distribution. |
|---|---|---|---|
| | AF2 | 1E-5 – 0.0001 | |
| | AF3 | 0.0001 - 0.0.001 | |
| | AF4 | 0.001 – 0.01 | |
| | AF5 | 0.01 – 0.1 | |
| | AF6 | 0.1 - 1 | |
| AF VZ | AF1 | 0 - 1E-5 | Attenuation factor during vertical solute transport through the parent material/vadose zone (see details above). $$AF\_VZ = \exp\left(\frac{-\ln(2) * RF\_VZ * max(Depth - dA - dB, 0) * FC\_VZ}{\frac{IR}{30} * HalfLife}\right)$$ The thickness of the vadose zone is given by the 'Depth to groundwater' node (Depth) minus the thickness of the topsoil and subsoil. It is assumed that the half-life during transport through the vadose zone is 1000 times longer than in the A horizon. Discretisation boundaries were based on expert opinion; a logarithmic scale was used to reflect the skewed distribution. |
| | AF2 | 1E-5 – 0.0001 | |
| | AF3 | 0.0001 - 0.0.001 | |
| | AF4 | 0.001 – 0.01 | |
| | AF5 | 0.01 – 0.1 | |
| | AF6 | 0.1 - 1 | |
| Groundwater AF | Almost complete | 0 - 1E-5 | The combined attenuation factor for the soil horizons and the vadose zone describes the fraction of the pesticide applied at the surface that will eventually reach the groundwater: $GW\_AF = AF\_A * AF\_B * AF\_VZ$ Discretisation boundaries were based on expert opinion; a logarithmic scale was used to reflect the skewed distribution. |
| | Very high | 1E-5 – 0.0001 | |
| | High | 0.0001 - 0.0.001 | |
| | Moderate | 0.001 – 0.01 | |
| | Some | 0.01 – 0.1 | |
| | Limited | 0.1 - 1 | |
| Groundwater flux [kg ha⁻¹ yr⁻¹] | Low | 0-1.0E-5 | The pesticide amount leaching to groundwater is calculated from the groundwater attenuation factor (GW_AF) and the application rate (AR): $Leach = AR * GW\_AF$ Leaching to groundwater is considered high if the pesticide mass flux to groundwater exceeds 0.0001 kg ha⁻¹ yr⁻¹. If a mass flux of 0.0001 kg ha⁻¹ yr⁻¹ is mixed in the top 0.1 m of the groundwater, this will result in a concentration of 0.1 μg l⁻¹, which is the drinking water standard. |
| | Medium | 1.0E-5 - 0.0001 | |
| | High | 0.0001 - 7 | |
| Fraction remaining (f_decay) | Very high | < 0.1 | Fraction of pesticide that will remain following application and decay during a dry period before the first rainfall event. $$f_{decay} = \exp\left(-\frac{t_{ro} * \ln(2)}{DT50}\right)$$ where $t_{ro}$ [days] is the dry spell length. Discretisation was based on interpolation to ensure that conditional probabilities for combination of parent node states are meaningful. |
| | High | 0.1 - 0.3 | |
| | Medium | 0.3 – 0.5 | |
| | Low | 0.5 – 0.8 | |
| | Very low | >0.8 | |
| Overland flow attenuation [unit less] | Very high | 0 – 1E-5 | Overland flow attenuation factor. |
| | High | 1E-5 -1E-4 | |





| | Medium | 1E-4 – 0.001 | $AF_{of} = \exp\left(-\dfrac{t_{ro} * \ln(2)}{DT50}\right) * Slope/Distance * (1 - E_{buffer})$ |
|---|---|---|---|
| | Low | 0.001 – 0.005 | Discretisation was based on interpolation to ensure that conditional probabilities for combination of parent node states are meaningful. |
| | Very low | >0.005 | |
| Overland flow flux [kg ha$^{-1}$ yr$^{-1}$] | Low | 0-1E-5 | The pesticide amount reaching the reservoir with runoff is assumed to be a function of the application rate (AR), the overland attenuation factor ($AF_{of}$) and the fraction of overland flow to maintain pesticide mass balance: $OLR = AR * AF_{of} * (1 - f_{leach})$ The discretization is based on the same consideration as for the 'Groundwater flux' node |
| | Medium | 1E-5 - 0.0001 | |
| | High | >0.0001 | |
| Surface water risk [µg l$^{-1}$] | Low | <0.01 | Surface water risk is the sum of the groundwater and overland flow fluxes. Discretisation was based on predicted likely pesticide concentration in the reservoir. This was calculated by multiplying the combined fluxes by the total field area in the catchment ($A_c$=192 ha) and dividing by the water volume in the reservoir ($V_{res}$ = 938,700 m$^3$): $C_{sw} = (L_{gw} + L_{of}) * A_c/V_{res}$ The risk was considered high if the resulting concentration was likely to exceed the drinking water standard for pesticides 0.1 µg l$^{-1}$. |
| | Medium | 0.01-0.1 | |
| | High | > 0.1 | |




**Appendix B: Results for overland water risk for all pesticides under all scenarios.**

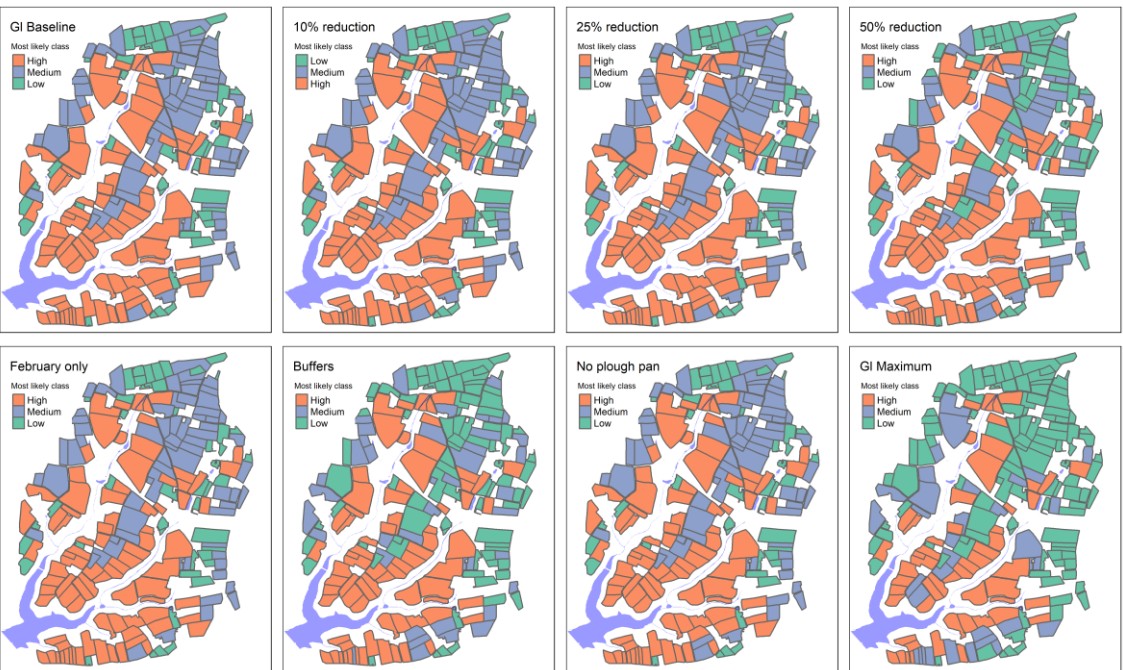

**Figure B1: Most likely overland flow risk class for Glyphosate under current application practices, reduction in application rate by 10%, 25% and 50%, shift of application to February, additional field buffers, no plough pan, and all mitigation measures combined.**





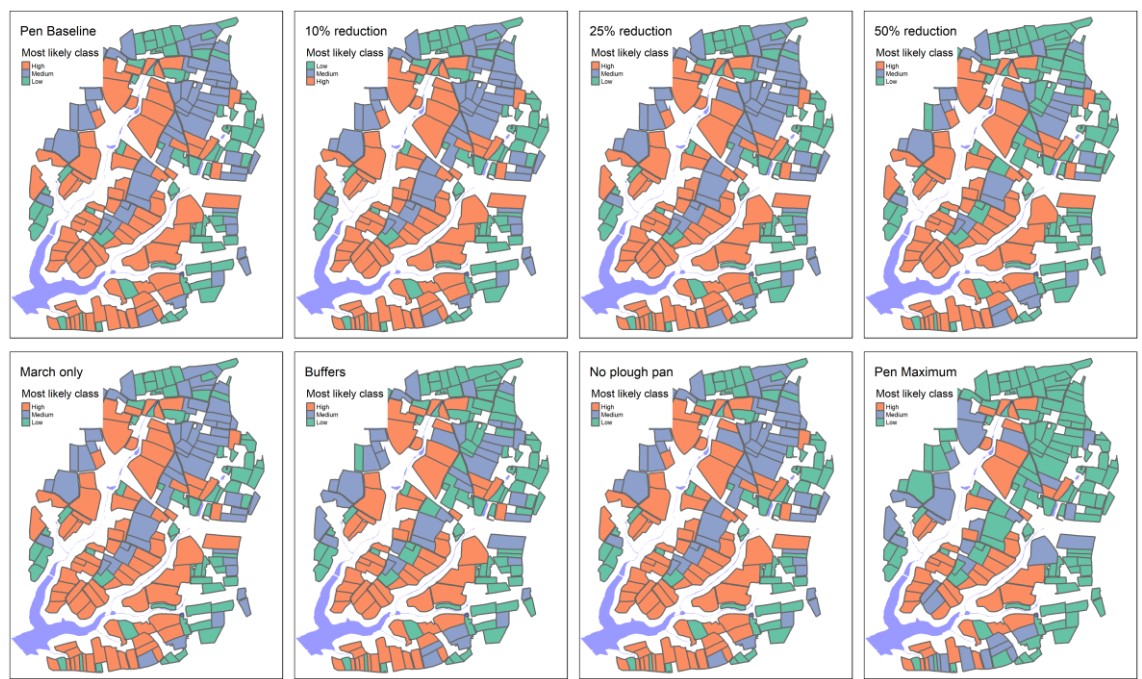

**Figure B2: Most likely overland flow risk class for Pendimethalin under current application practices, reduction in application rate by 10%, 25% and 50%, shift of application to March, additional field buffers, no plough pan, and all mitigation measures combined.**

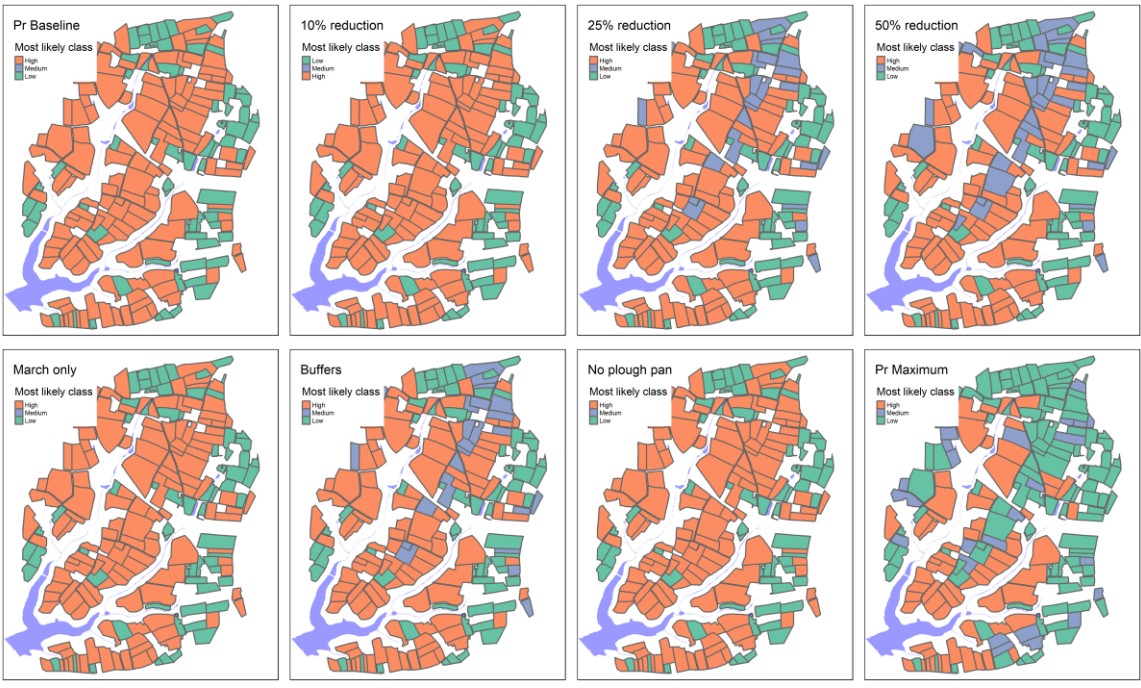

**Figure B3: Most likely overland flow risk class for Prosulfocarb under current application practices, reduction in application rate by 10%, 25% and 50%, shift of application to March, additional field buffers, no plough pan, and all mitigation measures combined.**



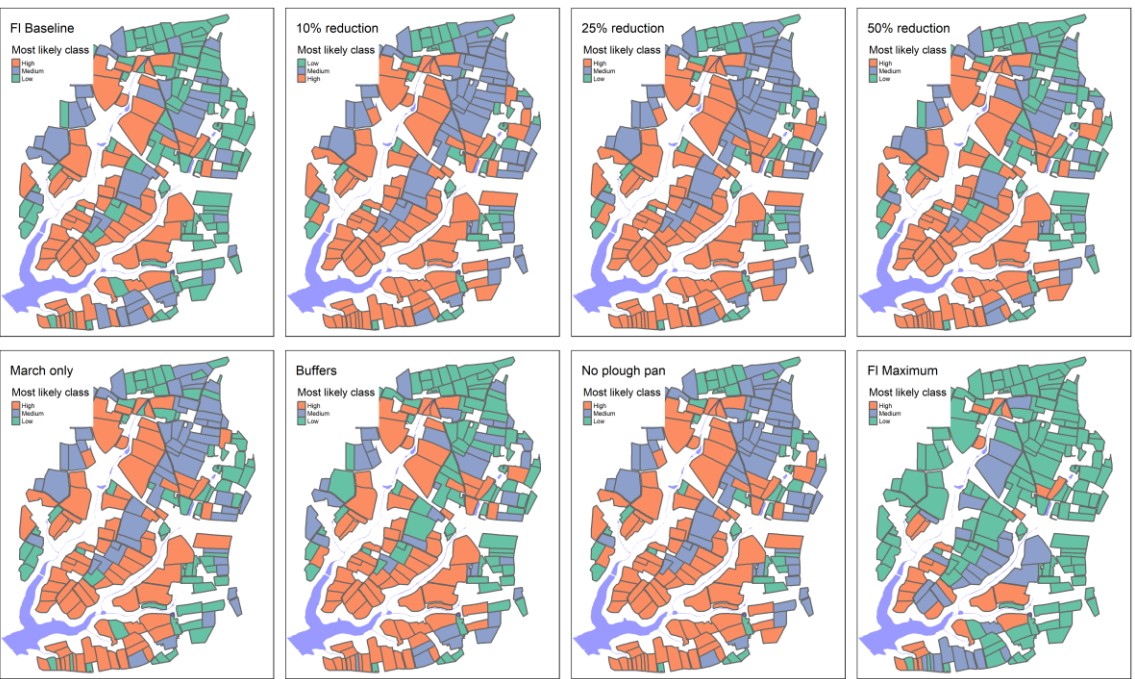

**Figure B4: Most likely overland flow risk class for Fluopyram under current application practices, reduction in application rate by 10%, 25% and 50%, shift of application to March, additional field buffers, no plough pan, and all mitigation measures combined.**


### Code and data availability

The code and data cannot not be made publicly available due to funder restrictions and privacy concerns. However, it may be available on request by contacting Director, Operations, Jersey Water, Mulcaster House, Westmount Road, St Helier, JE1 1DG, Jersey, UK.

### Author contributions

MG led conceptualisation, funding acquisition and project administration; MT, AL, ZG and AV undertook formal data analysis; MG and MT led model development and manuscript preparation; ZG led data curation and visualisation. All authors contributed to the methodological development, manuscript review & editing.





**Competing interests**

Authors have no competing interests to declare.

**Acknowledgements**

We thank Dr Jannicke Moe from the Norwegian Institute for Water Research for a detailed review of the manuscript. This research was funded by the Jersey Water Company Ltd. Further model refinement and manuscript preparation were funded by the Rural & Environment Science & Analytical Services Division of the Scottish Government.

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
