# Peer review of "Probabilistic modelling of inherent field-level pesticide pollution risk in a small drinking water catchment using spatial Bayesian Belief Networks"

_Hydrology and Earth System Sciences, 2021_

## Author Comment (AC2)

**Table 1: Summary of the pesticide monitoring data by location (P: Pump; E: East stream; W: West stream) in the VDLM catchment 2016-2019. Detection data are summarised as total number of samples, number of samples above limit of detection (LOD), and number of samples above the drinking water standard of 0.1 µg L$^{-1}$. The concentration data are summarised as mean ± standard deviation, minimum and maximum observed concentration.**

| Pesticide | Detection (total/ >LOD/ > 0.1 µg L$^{-1}$) | | | Concentrations (µg L$^{-1}$) (mean±std (min-max)) | | |
|---|---|---|---|---|---|---|
| | P | E | W | P | E | W |
| Glyphosate | 34/33/0 | 20/20/0 | 21/21/0 | 0.032 ± 0.018 (0.004-0.083) | 0.031 ± 0.026 (0.005-0.093) | 0.029 ± 0.030 (0.006-0.10) |
| Metobromuron | 27/27/27 | 4/4/4 | 6/6/6 | 0.215 ± 0.082 (0.10-0.40) | 0.200 ± 0.141 (0.10-0.40) | 0.733 ± 0.647 (0.20-1.70) |
| Pendimethalin | 258/107/0 | 129/42/0 | 245/122/5 | 0.010 ± 0.015 (<0.002-0.07) | 0.005 ± 0.004 (<0.002-0.02) | 0.018 ± 0.032 (<0.002-0.28) |
| Prosulfocarb | 67/55/14 | 6/3/1 | 6/6/3 | 0.098 ± 0.224 (<0.002-1.01) | 0.047 ± 0.104 (<0.002-0.26) | 0.318 ± 0.486 (<0.002-1.25) |
| Ethoprophos | 181/137/15 | 105/36/5 | 101/56/11 | 0.032 ± 0.046 (<0.002-0.27) | 0.014 ± 0.041 (<0.002-0.24) | 0.072 ± 0.277 (<0.002-2.43) |

---

## Author Response (AR1)

**RESPONSE TO REVIEWER 1 COMMENTS**

**This manuscript presents an interesting development of an hybrid Bayesian Belief Network, applied to the contamination risk of surface water by pesticides. This presents an interesting example of decision support tool with an implementation of a mix of different management options.**

**Though it is well written and nice to read, I have some concerns that need to be addressed before this manuscript gets published:**

We thank the reviewer for a positive assessment and for constructive comments. We have responded to the comments below and highlighted any proposed changes to the manuscript text using *italic*.

**Major concerns:**

**R1.1 - P7 l184-186: have you checked for time tendencies in climatology data. The mean values observed over the 1894-2019 period might be different from the 2016-2019 climate period on which the concentrations are observed.**
Thank you for the comment. We apologise for the imprecise specification of what was done in the model. We propose to amend this paragraph as follows:

*'Monthly rainfall and temperature data were available from a single meteorological station in the study catchment, operated by Jersey water for the period 2014-2019. Due to the short duration of this record, additional monthly total rainfall and mean monthly temperature data (from 1894-2019) were obtained from the Government of Jersey website (https://opendata.gov.je/organization/weather). Data for the years 1981-2019 were then used in combination with the catchment-based meteorological data to calculate monthly mean and standard deviation as priors for the model.'*

**R1.2: You describe your dataset as sparce. Could you please tell more about it, as it appears that the catchment on which you are working is rather well documented with many different sources of data.**
We appreciate the comment. The reviewer is correct that the modelling incorporates data from many different sources, and we believe we have already described and summarised this information in the manuscript (2.1-2.4) as well as in the appendix A. The reviewer is correct that the pesticide monitoring for some active ingredients (but not all) in the catchment by Jersey Water can be considered fairly substantial. Jersey Water currently scans for over 100 different pesticides using a risk-based water quality monitoring programme (cf. Jersey Waster's Annual Water Quality reports https://www.jerseywater.je/water-quality-report/). As suggested in Table 1, the number of pesticide observations available over the considered time period (2016-2019) vary for the different pesticides with pendimethalin and ethoprophos roughly being sampled on a weekly basis all year round, whereas prosulfocarb generally has been sampled on a weekly basis from mid-February to end of May, bi-weekly in January and otherwise every 3 to 5 weeks. The monitoring of glyphosate and metobromuron was more sporadic and only done during spring and early summer when these pesticides are expected to be used, with no sampling taken place in the autumn and the winter.

As described in section 2.2, data on pesticide application was obtained from Jersey Royals Company and are hence representative of actual application rates from Jersey island.

However, besides from the pesticide data, the data in the catchment is sparce. As described in section 2.3, there are very limited soils data available in the catchment, so we had to identify alternative sources, using Europe-wide soil property data derived from the HYPRES database. There is also very limited information available on hydrogeology and groundwater as well as on the catchment hydrology and water balance with the hydro-climatic data used being based on Jersey wide data. We had to use considerable ingenuity to identify alternative data sources and we are pleased that we seem to have succeeded – as the impression is of a 'data rich' catchment.

We suggest that we revise the first paragraph of section 2.2 to describe the pesticide monitoring in a bit more detail as follows:

'*Jersey Water, the sole water company and provider on Isle of Jersey, regularly tests and scans the quality of the raw water from the reservoir offtake for a range of different pesticides*. Five active pesticide ingredients currently or recently in use in the catchment showed evidence of significant concentrations in the reservoir offtake for the drinking water supply. These included the herbicides glyphosate, metobromuron, pendimethalin and prosulfocarb, and the nematicide and insecticide ethoprophos. *During 2016-2019 the sampling frequency of these pesticides was not the same (Table 1) with pendimethalin and ethoprophos being sampled on approximately a weekly basis all year round, whereas prosulfocarb generally was sampled on a weekly basis from mid-February to end of May, bi-weekly in January and otherwise every 3 to 5 weeks. The monitoring of glyphosate and metobromuron was more sporadic and only done during spring and early summer with no sampling taking place in the autumn and the winter*. Metobromuron was most frequently observed above the drinking water standard, followed by ethoprophos, prosulfocarb and pendimethalin (Table 1).'

**R1.3 - P10-13: the units in bracket along the text are not SI unit. Could you please check that and make the necessary changes, coherently with units used in Appendix A.**
Thank you for the comment. In the model development sections, we were describing and defining all model variables using SI dimensional units (L=length, M=mass, T=time etc.), a common way of describing generic units for inputs and parameters in models and equations, which avoids the need to introduce unit conversions directly in the equations. However, to avoid any confusion we will revise the method section 2.5 to use SI-units so that this section is consistent with the appendix and the results section as suggested by the reviewer.

**R1.4 - P10/11: there is an incoherence between equation (3) and Csw unit in microg.L-1. According to equation (3) which refers to fluxes, Csw should be a quantity per volume per year. Please check carefully your formulas and units to make it coherent.**
The reviewer is correct, the units in Eq. 3 are inconsistent. The reason for this inconsistency was that we were referring to annual pesticide application rates or fluxes (i.e., mass per area per time), when in fact we were actually considering the amount of pesticide mass applied per area in the model. We will change the term flux to load throughout the revised manuscript (and appendix A) and will make it clear that we refer to applied mass per area.

**R1.5: I have a concern about the way Csw is computed. Why did you divide the fluxes by the total volume of the reservoir. Doing so, you consider that a field is at risk only if its contribution to the reservoir contamination is enough to make it higher than the standard drinking limitation. This appears rather limiting the potential risk. And what about the sum of fields contribution to assess the catchment risk?**

Thank you for the comment. It is true that when calculating Csw, the flux (load) calculated from a given field (Lgw+Lof) is divided by the total volume of the reservoir, but please note that in Eq. 3 we also multiply the load by the total area of ALL fields in the catchment (and not by the area of the given field). As stated after Eq. 3, this calculation essentially corresponds to assuming that the load from every field in the catchment is the same as the load computed for the given field. Please note that this calculation was primarily done to help define the discretisation boundaries (low, medium, high) for the combined surface water risk node. We suggest rephrasing how Csw is calculated to make it clearer that we use the total area of all fields in the catchment for the calculations as follows (please also see our response to comment 2 by the editor):

'This combined pesticide load was converted to a surface water concentration $C_{sw}$ [µg L$^{-1}$] to evaluate the risk to surface water as follows:

$$C_{sw} = (L_{gw} + L_{of}) * \frac{A_c}{V_{res}} * 10^6 \qquad (3)$$

where $A_c$ is the total area of all fields in the catchment (192 ha) and $V_{res}$ is the water volume in the reservoir (938,700 m3). Equation 3 is a very simplified way of evaluating the field-level risk; it is essentially assumed that the combined load from the field in question represents the average load from all fields in catchment. This allows the combined pesticide load from the given field to be converted to a concentration in the reservoir, which can then be compared to the regulatory standards based on which the risk can subsequently be assessed. Hence, if the combined pesticide load from a field resulted in $C_{sw}$ exceeding the standard of 0.1 µg L-1, this field was considered high risk (see Appendix A).'*

As the reviewer points out, the loads/contributions from each field could be calculated and summed to assess the overall catchment risk. As discussed in section 3.5 (we propose to move this to section 3.4 in the revised version of the manuscript as stated in response to **R1.8** below), the spatial application in the R package bnspatial allows to simulate expected quantities, based on the median value of each discretisation interval. Hence, by multiplying the combined loads from each field and summing the resulting pesticide masses over all fields in the catchment and dividing by the reservoir volume, a concentration in the reservoir water could be estimated for each pesticide and month, which could be compared to measured concentrations if available for further model validation. However, we found that this deterministic calculation was heavily reliant on the discretisation of the target node in question, and, coupled with rare extreme high values generated by the stochastic model, made such "validation" uncertain. We therefore did not follow this approach. We will revise section 3.4 on model validation to make this aspect clearer as follows:

*'A qualitative 'reasonable fit' visual inspection has been shown to be an effective means of assessing model performance using diverse incomplete data sets (Ghahramani et al., 2020). Ghahramani et al. (2020) found that the ranking of confidence in model predictions between determinands was related to data availability as much as to the model itself, with pesticide simulations performing less well than those for hydrology, sediments and phosphorus.*

*Further validation approaches could be explored in future implementations. The spatial application in the R package bnspatial allows to simulate expected quantities, based on the median value of each discretisation interval. Hence, by multiplying the expected loads from each field with the probability of the field falling into each discretised interval, then summing the resulting pesticide masses over all fields in the catchment and dividing by the reservoir volume, a concentration in the reservoir water could be estimated for each pesticide and month. This could then be compared to measured concentrations if available for further model validation. However, this deterministic calculation would be heavily reliant on the discretisation of the target node in question and, coupled with rare extreme high values generated by the stochastic model, would make the validation uncertain. Hence, it would be best applied in combination with dynamic discretisation, if available, and with further model development constraining upper simulated values.'*

**R.1.6 - P15l416-417: what volume did you use as a dmix for the overland flow flux (comparatively to eq 11)?**
We assumed the same mixing volume for the overland flow load as used for the groundwater load. We did this purely to allow a comparison of the relative contribution of the two components to the combined risk. Defining the discretisation boundaries (low, medium, high) for the groundwater load was based on an assumption that the pesticide load $L_{gw}$ reaching the groundwater was mixed in the upper $d_{mix}=10$ cm of the groundwater aquifer; the resulting concentration ($C_{gw} = L_{gw}/d_{mix}$) can then be compared to regulatory standards based on which the discretisation boundaries were defined. It is therefore relatively straightforward to define the boundaries for the groundwater load; all that is needed is a mixing depth assumption. It is harder to do the same for the overland flow load, as it is not easy to define the depth (or volume) that the load (or mass) should be mixed in. For simplicity, we therefore opted for using the same boundaries as for groundwater load.

However, following comment 1 by the editor we have now removed the "mixing" aspect from the method section in the manuscript text (i.e., we deleted L367-375 and L416-417 in the original version of the manuscript). Please see response to comment 1 by the editor.

**R1.7: Have you performed any inference according to observed concentrations data. If so, please give details about it, if not, please state clearly your model remains an expert based model.**
No, we did not perform any inference according to observed concentration data. We will emphasise this in section 3.4 of the revised manuscript that our model is an 'expert-based' model.

**R1.8: The results/discussion sections are merged. This could be acceptable, but the subsections 3.2/3.3/3.4 are only results with no confrontation to literature or prior hypothesis. This would be great to add discussion elements to these paragraphs or to move them to an independent result section.**
Thank you for this point. We will put our findings in a wider context by discussing them in relation to literature and propose to amend the text as follows.

In section 3.2 we proposed to include additional text:

*'While the above influential variables have been previously identified among important factors for rapid overland flow risk (Bereswill et al., 2014; Tang et al., 2012), our modelling approach allows to distinguish between generic risk factors and variables with the greatest influence on pesticide pollution*

*risk in a local catchment setting. Coupled with the ability to identify critical source areas, and how they change at a monthly time step, our approach offers a major advancement as compared to static GIS-based risk index assessment approaches (e.g. Quaglia et al., 2019).'*

In section 3.3 we propose to revise the text to:

'Reduction in application rates unsurprisingly results in a reduced risk, particularly in the groundwater leaching risk (Figure 9), which is reduced to Low even after 10% reduction. *This is in line with findings of Reichengerger et al. (2007) who concluded that application rate reduction was one of a few mitigation measures that could address pesticide leaching to groundwater with some confidence. However, while reduction in application rates is an easy measure to implement and may lead to cost savings, it can only be implemented to a point whereby it remains effective, thus potentially limiting its acceptability as a mitigation measure to farmers (Bereswill et al., 2014).* Introduction of buffers reduces the runoff risk to a similar extent as a 50%  reduction of pesticide application rates for overland flow risk. *Buffers, particularly edge of field buffers, have been found to be effective mitigation measures for the overland flow pathway (Bereswill et al., 2014; Reichenberger et al., 2007). Although buffer effectiveness is variable, most studies report efficiencies over 60% (Bereswill et al., 2014).In addition, permanent buffers are seen as an acceptable mitigation intervention, as long as farmers can be compensated for potential losses of growing area (Bereswill et al., 2014).* Managing and removing potential plough pans increases the amount of infiltration into soils, thus reducing the pesticide runoff risk to an extent that is comparable to 10% reduction in application rates (Fig. 8). By combining all available maximum interventions of 50% reduction in pesticide application rate, management of plough pan, delayed application timing and installing additional field buffers, the probability of all types of risk is notably reduced (Figs. 4-6, 8-9). *Whilst some of these measures may be deemeed less acceptable or difficult to implement (e.g. delayed application timing (Bereswill et al., 2014)), an ambitious mitigation approach may be required to achieve real water quality improvements (Villamizar et al., 2020).'*

In section 3.4 we propose to revise the text (and move some of the text from 3.5) to:

*'We constructed a causal network, where the model structure was informed by expert knowledge. However, Bayesian Networks can also be used as machine-learning associative tools that are suitable for deriving patterns in datasets without a specific response variable. It could be argued that pesticide risk, expressed as load or concentration of pesticide in different potential loss pathways (overland flow or groundwater leaching) is a latent variable without available observational data (Piffady et al., 2020), making it difficult to calibrate or validate a risk model. Hence, model credibility and salience (Cash et al., 2005) need to be evaluated by experts and stakeholders. Here, we implemented a simple validation approach to confirm that the model predictions fall within the realms of credibility, using the limited observational data to validate the expert-based model.*

Figure 10 shows a comparison between the probability density distributions based on 10,000 surface water risk simulations for each active ingredient in the hybrid network and the limited observational data (in µg L⁻1) available for the months January – March between 2016 and 2019. The model typically over-estimates the simulated risk for glyphosate and pendimethalin, albeit with low probability of high values. The simulated and observed distributions for prosulforcarb are comparable, whilst the model seems to under-estimate the risk from metobromuron. However, it has to be noted that the very few observations available for metobromuron (N=8) seem to be higher and less accurate than for the other pesticides. It should also be noted that the developed model was never intended to represent the complex transport and fate processes in the catchment in detail or to accurately simulate the pesticide concentration levels in the reservoir, so the comparison in Figure 10 was mainly

carried out as a sense check of the model predictions. Overall, model simulations appear conservative, which is helpful in terms of informing a precautionary management approach. *Further model refinement could focus on constraining the upper simulation values throughout the model. A qualitative 'reasonable fit' visual inspection has been shown to be an effective means of assessing model performance using diverse incomplete data sets (Ghahramani et al., 2020). Ghahramani et al. (2020) found that the ranking of confidence in model predictions between determinands was related to data availability as much as to the model itself, with pesticide simulations performing less well than those for hydrology, sediments and phosphorus.*

*Further validation approaches could be explored in future implementations. The spatial application in the R package bnspatial allows to simulate expected quantities, based on the median value of each discretisation interval. Hence, by multiplying the expected loads from each field with the probability of the field falling into each discretised interval, then summing the resulting pesticide masses over all fields in the catchment and dividing by the reservoir volume, a concentration in the reservoir water could be estimated for each pesticide and month. This could then be compared to measured concentrations if available for further model validation. However, this deterministic calculation would be heavily reliant on the discretisation of the target node in question and, coupled with rare extreme high values generated by the stochastic model, would make the validation uncertain. Hence, it would be best applied in combination with dynamic discretisation, if available, and with further model development constraining upper simulated values.'*

**R1.9 - P16 l456-458 and Section 3.4: have you simulated 10.000 results for each field? It is unclear here. And once again, I have a concern with the direct comparison of Csw and concentrations as concentrations are the sums of all field contributions. Furthermore, it might be interesting to also consider / discuss the fact that ground fluxes are probably slower than surface ones.**
Thank you for the comment. For the validation, we did not run 10.000 simulations for each field but instead tested the simulations derived from the hybrid model against observational data, before discretisation and spatial application was implemented. We will clarify this in the text (L637) as:

'Figure 10 shows a comparison between the probability density distributions based on 10,000 surface water risk simulations for each active ingredient in the hybrid network and the limited observational data'.

Please also see our response to comment 3 by the editor regarding the 10,000 simulations and our revised section 2.5.3 for more clarification.

Please refer to our response regarding the calculation of Csw in R1.4 above. The reviewer is correct, the groundwater fluxes will be slower than the runoff ones. This is something the model is currently not accounting for. We will highlight this in the limitation and outlook section 3.5 as:

'Although quite detailed in process representation that is based on established mechanistic approaches, the modelling of pesticide leaching and runoff in the BBN is still simplified and could be extended to e.g., consider preferential flow pathways *and/or capture the slower rate of groundwater pesticide leaching as compared to fluxes via surface runoff.*'

**Minor comments and typos:**

**R1.10 - P5 l142: why is fluopyram underlined?**
Thank you for pointing out this error, which will be corrected in the revised manuscript

**R1.11 - P11 l320: check reference brackets.**
We will correct this formatting error.

**R1.12 - P15 l419: typo on 'asses'.**

We will correct this typographical error.

**R1.13 - P16 l471: Piffady (2020) approach was clearly not mechanistic. This would be more helpful to cite here an example of mechanistic approach.**

Thank you for the comment. We will remove the reference to Piffady and replace the reference with the studies by Vilamizar et al. 2020 (using SWAT to model pesticide and influence of management interventions) as well as the review paper by Kohne et al. (2009) focussing on mechanistic modelling of pesticide transport at various scales.

**R1.14 - P24 l578-580: a reference would be needed.**

Thank you for spotting the missing reference. We will add the reference to the study by Reichenberger 2007 in the revised manuscript.

**R1.15 - P24 l581: typo on 'metabrouron'. Please check metabromuron spelling through the document, you sometimes write it with an h, sometimes without.**

Thank you for spotting this inconsistency. We will check the spelling and ensure that we write 'metobromuron' throughout the revised manuscript

**R1.16 - P25 l584: please precise which figure you refer to.**

Apologies for the omission, we will amend the Figure number in the revised manuscript.

**R1.17 - P29 l666: '..' typo.**

We will replace the … with (…) to mark the gap in the text quoted from the Schuwirth et al. (2019) page 2 and add the page number in the manuscript

**R1.18 - P32 actual evapotranspiration: there are overlapping boundaries between low and medium classes.**

Thank you for spotting this error. This will be corrected as Low <35, Medium 35-40, High >40

**R1.19 - P35 'groundwater fluxes': I would expect an 'f-leach' factor in the Leach formula, coherently with eq (1).**

The reviewer is correct. We will correct this in a revised manuscript

**R1.20 - P36 'surface water risk' : according to my units checking, there should be a time factor in the unit (see major concerns).**

Please see the response to major comment in **R1.4.**

**RESPONSE TO REVIEWER 2 COMMENTS**

**This study presents an innovative and interesting work on probabilistic modelling of pesticide pollution risk in Jersey Island. The manuscript was well written and in my opinion, this manuscript can be accepted after a few minor corrections.**

We thank the reviewer for the positive feedback and constructive comments. We have replied to each of the comments below and highlighted any proposed changes to the manuscript text using *italic*.

**Specific Comments:**

**R2.1 - L125, Page 5: "µg l-1" should be replaced as "µg L-1". Same corrections need to be done for Table 1.**
We will change the symbol for litre from l to L consistently throughout the revised manuscript.

**R2.2 - Table 1: The author has provided the mean concentrations for five pesticides along with the min and max limit. However, standard deviation needs to be included for each of the pesticides to make the table more statistically robust.**
We will revise Table 1 to also include standard deviation of the pesticide concentration data. We propose to revise Table 1 and table caption as shown below:

Table 1: Summary of the pesticide monitoring data by location (P: Pump; E: East stream; W: West stream) in the VDLM catchment 2016-2019. Detection data are summarised as total number of samples, number of samples above limit of detection (LOD), and number of samples above the drinking water standard of 0.1 µg L$^{-1}$. The concentration data are summarised as mean $\pm$ standard deviation, minimum and maximum observed concentration.

| | Detection (total/ >LOD/ > 0.1 µg L$^{-1}$) | | | Concentrations (µg L$^{-1}$) (mean±std (min-max)) | | |
|---|---|---|---|---|---|---|
| **Pesticide** | **P** | **E** | **W** | **P** | **E** | **W** |
| Glyphosate | 34/33/0 | 20/20/0 | 21/21/0 | 0.032 $\pm$ 0.018 (0.004-0.083) | 0.031 $\pm$ 0.026 (0.005-0.093) | 0.029 $\pm$ 0.030 (0.006-0.10) |
| Metobromuron | 27/27/27 | 4/4/4 | 6/6/6 | 0.215 $\pm$ 0.082 (0.10-0.40) | 0.200 $\pm$ 0.141 (0.10-0.40) | 0.733 $\pm$ 0.647 (0.20-1.70) |
| Pendimethalin | 258/107/0 | 129/42/0 | 245/122/5 | 0.010 $\pm$ 0.015 (<0.002-0.07) | 0.005 $\pm$ 0.004 (<0.002-0.02) | 0.018 $\pm$ 0.032 (<0.002-0.28) |
| Prosulfocarb | 67/55/14 | 6/3/1 | 6/6/3 | 0.098 $\pm$ 0.224 (<0.002-1.01) | 0.047 $\pm$ 0.104 (<0.002-0.26) | 0.318 $\pm$ 0.486 (<0.002-1.25) |
| Ethoprophos | 181/137/15 | 105/36/5 | 101/56/11 | 0.032 $\pm$ 0.046 (<0.002-0.27) | 0.014 $\pm$ 0.041 (<0.002-0.24) | 0.072 $\pm$ 0.277 (<0.002-2.43) |

**R2.3 - L140, Page 5: Remove the underline from "fluopyram".**
Thank you for spotting this error, we will correct it.

**R2.4 - Since, fluopyram was considered for the final model instead of ethoprophos, concentration, and detection level of fluopyram need to be included in table 1.**
There are no monitoring data available for fluopyram in the catchment, as fluopyram is currently not widely used in the study catchment. As explained in P5 L141, fluopyram was included in the model as a potential future replacement of ethoprophos. Ethoprophos has previously been widely used in

the catchment, but it has now been banned and its use is therefore discontinued. This will be clarified in the revised manuscript as follows:

'Five active pesticide ingredients currently or recently in use in the catchment showed evidence of significant concentrations in the reservoir offtake for the drinking water supply. These included the herbicides glyphosate, metobromuron, pendimethalin and prosulfocarb, and the nematicide and insecticide ethoprophos. Metobromuron was most frequently observed above the drinking water standard, followed by ethoprophos, prosulfocarb and pendimethalin (Table 1). Ethoprophos was not included in the final model, as its use has now been discontinued. *In consultation with Jersey Water, it was instead decided to include the nematicide fluopyram as it was considered to be a likely replacement for ethoprophos and hence a potential future risk to the reservoir water quality. Monitoring data for fluopyram are not available as it is currently not widely used in the catchment. However, fluopyram is known to be used at lower application rates than ethoprophos, making its potential risk of contaminating the water supply intrinsically lower, notwithstanding its relatively high mobility and greater persistence.'*

**R2.5 Page 10, Equation 3: The author has mentioned, "combined pesticide flux" which was converted to surface water concentration to evaluate the risk. However, there is some inconsistency in the units. Not clear. please check equation 3.**
The reviewer is correct, the units in Eq. 3 are inconsistent. The reason for this inconsistency was that we were referring to pesticide application rates or fluxes (i.e., mass per area per time), when in fact we were actually considering the amount of pesticide mass applied per area. We will change the term flux to load throughout the revised manuscript and make it clear that we refer to mass per area, as also outlined in reply to **R1.4**.

**R2.6 - L415, Page 15: Is there any specific volume for the overland flow flux?**
To assess whether a calculated overland flow load should be classified as low, medium or high, we used the same discretisation boundaries as we used for the groundwater load. We did this purely to allow a comparison of the relative contribution of the two components to the combined risk. Defining the discretisation boundaries for the groundwater load was based on an assumption that the pesticide load $L_{gw}$ reaching the groundwater was mixed in the upper $d_{mix}$=0.1 m of the groundwater aquifer; the resulting concentration ($C_{gw} = L_{gw}/d_{mix}$) can then be compared to regulatory standards based on which the discretisation boundaries were defined. It is therefore relatively straightforward to define the boundaries for the groundwater load; all that is needed is a mixing depth assumption. It is harder to do the same for the overland flow load, as it is not easy to define the depth (or volume) that the load (or mass) should be mixed in. We therefore opted for using the same boundaries as for groundwater load. Essentially this means the mixing volume is the same for the overland flow load as for the groundwater load.

However, following comment 1 by the editor we have now removed the "mixing" aspect from the method section in the manuscript text (i.e., we deleted L367-375 and L416-417 in the original version of the manuscript). Please see our response to comment 1 by the editor for further clarification.

**R2.7 - ð••»(ð•'‹) = −∑ð•' $f$(ð•'‹) log2(P(X)) should be equation 16?**
We will follow this suggestion and name the entropy equation as equation 16. Note that because Equation 11 has been removed in the revised manuscript, this is now Equation 15.

**R2.8: Author has mentioned the soil properties. It will be great to have that information in a tabular format. Specifically the soil organic carbon concentration and pH data.**

Thank you for the comment. All the soil properties used in the model (bulk density, field capacity, organic carbon, and hydraulic conductivity) are tabulated in Appendix A. These data were all taken from the HYPRES data base. There are only limited soil property data (pH and SOM) available in the VDLM catchment and we decided to summarise these in the text (page 9) rather than in a tabular format given these data were not used in the modelling in the end. As stated in the text on line 239 of the original manuscript, we had topsoil SOM data for 40 out of the 200 fields, and pH for 37 fields. We considered using the median or mean SOM per hydrogeological unit and per crop to extrapolate to all fields, however we had no observed SOM data for fields on blown sand. Therefore, for consistency, we decided to use data from the HYPRES database, as this also provided SOM estimates for the lower soil horizons as well other soil parameters (bulk density, field capacity and hydraulic conductivity) needed for the model. For comparison, we will include summary statistics (mean, min, max) for the observed SOM data for loess and shale in the text on line 241.

*'Mean observed SOM for loess was 2.5% (1.9-3.8%) and 2.3% for shale (1.9-3.4%).'*

**R2.9 - Page 25: "particularly in the groundwater leaching risk (Figure ??"): Please mention the figure number.**

Thank you for spotting the missing figure number. This will be amended in the revised manuscript as Figure 9.

**R2.10: There are quite a number of typographical errors. Please review the manuscript carefully.**

We apologise for typographical errors and will carefully review the manuscript to remove any errors from the revision.

**RESPONSE TO EDITOR COMMENTS**

**EDITOR COMMENTS**

**I am generally content with the responses to the valuable comments received from the two reviewers. I have a few comments/suggestions that I hope that you will address prior to the publication of the paper. Some of the comments are follow-ups on the comments that were raised by the reviewers. The comments are listed below:**

We thank the editor for the positive feedback and constructive comments. We have replied to each of the comments below and highlighted any proposed changes to the manuscript text using italics.

**1) The response to Reviewer's 1 "R.1.6 - P15l416-417: what volume did you use as a dmix for the overland flow flux (comparatively to eq 11)?" is not convincing. Dividing overland flow with dmix does not make much sense as overland flow is largely moving as sheet flow. I also do not understand the need to divide with dmix. This point was also mentioned by reviewer 2.**

We appreciate the comment. We understand that the dmix is causing confusion and we therefore propose to remove this aspect from the method section in the manuscript text (i.e., delete L367-375 and L416-417). The hybrid BBN model calculates the pesticide loads to groundwater and to the reservoir via overland flow, respectively, and these calculations do NOT depend on dmix. We only used the dmix assumption to help define the discretisation intervals (low, medium, high) of the groundwater load and overland flow load nodes in the discrete (spatial) version of the BBN.

For the groundwater load node, we are currently assuming the following discretisation in the spatial (discretised) BBN:
Low: < 0.00001 kg/ha
Medium: 0.00001 – 0.0001 kg/ha
High: > 0.0001 kg/ha

The justification for this discretisation is that a load larger than 0.0001 kg ha-1 would result in a concentration > 0.1 ug/l if mixed in the upper 10 cm of an aquifer, and hence is considered high.

For simplicity and to allow direct comparison of the groundwater load and the overland flow load, we decided to use the same discretisation intervals for the overland flow. We acknowledge that we hereby implicitly use the same mixing depth assumption as for the groundwater load to define the discretisation intervals for the overland flow load and that this from a physical point of view makes less sense. However, we still find that this approach allows for a "fairer" direct comparison of the two load components.

However, we will accept if the editor does not agree with this/find the argument unconvincing, in which case we will rethink how to discretise the overland flow node. An alternative option would for example be to divide the overland flow pesticide load with the overland flow rate to get the concentration in the water that reaches the reservoir via overland flow – and use this concentration as basis for the discretisation. A similar approach could be used for groundwater load by dividing the load with the infiltration rate.

**2) Equation 3 in the response to reviewer 1 should have a summation over all fields.**

We appreciate the comment, however Eq. 3 should NOT have summation over all fields. What we are attempting to do here is to calculate the spatial pesticide risk, i.e. we want to assign a risk class to each field in the catchment. At the moment this risk calculation is based on the sum of the groundwater pesticide load (Lgw) and the overland flow load (Lof) from the given field. Similar to the discussion in comment 1 above, the question is how to decide what values of the field-level combined load (Lgw+Lof) should be considered 'risky' or 'non-risky'.

The way we have done this is as follows:

We know the volume of the reservoir $V_{res}$ (938,700 m3). We can therefore calculate the total pesticide mass $M_t$ required for the whole reservoir to exceed the drinking water standard of 0.1 mg/m3, i.e.:

$$M_t = C_{standard} * Vr_{es} = 0.1 \text{ mg/m3} * 938,700 \text{ m3} * 10^{-6} \text{ kg/mg} = 0.094 \text{ kg}$$

By dividing this mass by the total area of the catchment $A_c$ (192 ha), we can determine the catchment-average load (kg/ha) that will result in the drinking water standard being exceeded, i.e.:

"Risky" load to reservoir (kg/ha) = $M_t/A_c = C_{standard} * V_{res} /A_c = 0.00049$ kg/ha.

We then say that any individual field that has a combined load (Lgw+Lof) larger than this "risky" load is classified as high risk.

We believe this is a meaningful and fair basis for evaluating whether the combined load from an individual field is risky or not (low, medium or high).

The above also means that rather than expressing the final risk node/results as the combined load (Lgw+Lof) from a field, we can express it as a surface water concentration, which is exactly what was shown in Eq. 3. Expressing the final node as a surface water concentration in the model makes it easier to compare with regulatory standards and/or observed concentrations directly in the model.

We suggest modifying the text after Eq. 3 to make it clearer that in order to evaluate the field-scale risk, we are effectively assuming that the load from the field in question represents the average load from all fields in the catchment as follows:

*where $A_c$ is the total area of all fields in the catchment (192 ha) and $V_{res}$ is the water volume in the reservoir (938,700 m3). Equation 3 is a very simplified way of evaluating the field-level risk; it is essentially assumed that the combined load from the field in question represents the average load from all fields in catchment. This allows the combined pesticide load from the given field to be converted to a concentration in the reservoir, which can then be compared to the regulatory standards based on which the risk can subsequently be assessed. Hence, if the combined pesticide load from a field resulted in $C_{sw}$ exceeding the standard of 0.1 µg L-1, this field was considered high risk (see Appendix A).*

**3) How and why the 10,000 simulations were conducted needs to be clarified further. This point was raised by Reviewer 1 R1.9 - P16 l456-458 and Section 3.4.**

The 10,000 random simulations for each pesticide were generated in the hybrid BBN using forward stochastic (Monte Carlo) simulation with the samples being drawn according to the nodes' prior probability distributions. GeNie's forward stochastic sampling algorithm is very efficient for hybrid

models in cases when no evidence has been observed or when the network contains "hard" evidence for parentless nodes only. Because the prior probability distributions of the input nodes in the hybrid model reflect the catchment as a whole (for example, the prior distribution of crop type represent the actual distribution of the crop types in the catchment and so on), the simulated surface water concentrations will also to some extent be representative of the catchment as a whole. Obviously, the simulated results with the hybrid model do not account for spatial patterns and potential correlation in the inputs, in the same way as is accounted for in the spatial application of the model using bnspatial, where the model is applied on a field-by-field basis.

We found that 10,000 simulations were appropriate for achieving stable results for the target nodes. The plot below shows an example of the mean and standard deviation of the simulated concentration (surface water risk node) with the hybrid BBN as function of number of simulations. This shows that both the mean and the standard deviation is stable after 5000+ simulations and confirms that 10000 simulations are adequate.

To clarify the above we suggest revising the last paragraph of section 2.5.3 the manuscript as follows:

*For model validation, simulated values of surface water risk (i.e., surface water pesticide concentration in $\mu g\ L^{-1}$) in the hybrid model were compared to the limited available water quality observations for four active ingredients from month January – March (see Section 3.4). This was done by generating 10,000 random simulations for each pesticide in the hybrid BBN using forward stochastic simulation, with the random samples being drawn from the nodes' prior probability distributions. It was found that 10,000 simulations were adequate for achieving stable model convergence. Note that in the hybrid model the prior probability distributions for the different nodes reflect the catchment as a whole (e.g., the prior probability distribution for the crop type node reflects the actual distribution of crop types in the catchment). This means that although spatial patterns and potential spatial correlation between inputs are not accounted for explicitly in the hybrid model (unlike for the spatial application of the model using bnspatial, where the model is applied on a field-by-field basis), the simulated surface water concentrations in the hybrid model, , can be considered representative of the average catchment conditions.*

[Figure]

**4) Table 1. Prefer that you report medians instead of means given that you have many below detection values.**

Ok we have revised Table 1 to include medians rather than means.

**5) Line 204: the authors used Topographic connectivity that is derived by calculating the horizontal distance from the polygon vertex nearest to the stream features using the Distance to nearest hub tool in QGIS. Why not use connectivity based on actual flow direction? That is more accurate and accounts for water movement.**

We thank the editor for the comment. The editor is correct, the connectivity of each field was based on the calculation of the horizontal distance from the polygon vertex nearest to the stream network. We opted for this simpler approach because the study catchment is flat with relatively uniform slopes (meaning there was little difference between distance to stream and flow accumulation). While we could have used the polygon centroid, we selected the polygon vertex closest to the stream as a precautionary approach.

Using the distance to stream approach has also previously been shown to be accurate for assessing connectivity (e.g., Grauso, S., Pasanisi, F., Tebano, C., 2018. Assessment of a Simplified Connectivity Index and Specific Sediment Potential in River Basins by Means of Geomorphometric Tools. Geosciences 8, 48, https://doi.org/10.3390/geosciences8020048). The distance to stream approach was also used in relatively flat cultivated catchments in Scotland for P-risk applications with good results (Gagkas, Z., Lilly, A., Stutter, A., Baggaley, N. 2019, Development of framework for a Red-Amber-Green assessment on phosphorus application to land. Report prepared for SEPA).

We did check the flow direction and flow accumulation in the VDLM catchment and found that total diffuse flow (instead of convergent flow) was the dominant one in the study fields, which justified not looking at flow accumulation for assigning connectivity. However, we agree that in a more diverse landscape it would be better to use a connectivity index based on flow direction and accumulation.

We suggest clarifying the above considerations by adding the following:

*Because the study catchment is flat with relatively uniform slopes, we opted here to base the topographic connectivity on the simple distance-to-stream approach. This approach has previously been shown to be accurate for assessing connectivity (e.g., Grauso et a., 2018). For other catchments with a more diverse topography, it might be more appropriate to use a connectivity based on actual flow direction.*

**6) Line 284: AFgw and AFof are introduced but are not defined. Please refer to their respective equations in the paper to allow readers to better understand these two important parameters.**
Thank you, we will add reference to the AF equations by rewriting L284 as follows:

*During transport to groundwater and surface water the pesticide can undergo attenuation with the degree of attenuation determined by attenuation factors, respectively AFgw (Eq. 10) and AFof (Eq. 12) (as described in the following sections).*

**7) Line 354: The equation for AFi. Indicate that i ranges between 1 and 3.**
Thank you for the comment. Rather than adding that i ranges from 1 to 3 in Eq. 8 (and Eq. 9), we will clarify that the subscript i refers to zones A, B or VZ by adding the following before Eq. 8:

*An attenuation factor can be calculated for each of the three zones $AF_i$ (where subscript i refers to the A horizon (A), B horizon (B) or the vadose zone (VZ)):*

**8) Lines 400-405: Please provide support for the selected cutoffs values for f_slope and f_dist?**
The cutoff for f_slope is very similar to the assumed cutoff in REXTOX (as well as in e.g. Drewry, J. J., Newham, L. T. H. and Greene, R. S. B.: Index models to evaluate the risk of phosphorus and nitrogen loss at catchment scales, J. Environ. Manage., 92(3), doi:10.1016/j.jenvman.2010.10.001, 2011). The distance correction and cut-off value was based on the assumption that no significant pesticide attenuation would occur within 1m of a watercourse. This assumption was informed by (Reichenberger et al., 2007) who state that:' …even if pesticide-loaded runoff infiltrates into a riparian buffer strip, the groundwater table below the strip will be rather shallow (unless the stream bed is deeply cut into the floodplain), and the groundwater feeds into the nearby stream… we conclude that riparian buffer strips are most probably much less effective than edge-of-field buffer strips.' This has now been clarified in the manuscript.

**9) Lines 412-413: The sentence "It should be noted that REXTOX only considered pesticide losses via runoff from fields adjacent to surface waters and therefore did not include the effect of distance from field to water body" is not clear. Please revise.**
We will revise this to make it clearer that REXTOX only consider fields immediately adjacent to surface waters and therefore does not include a distance correction factor.

**10) Provide more information on how the SHELF model was used to generate the prior probability distributions for the continuous nodes.**

We proposed to expand the text in the manuscript as follows:

*Prior probability distributions for continuous nodes were fitted to available data using the min, max, 5th, 50th and 95th percentiles of the cumulative probability distribution (O'Hagan, 2012) in the SHELF package (Oakley, 2020) in the open-source statistical modelling software R (The R Project for Statistical Computing 4.0.1). In the SHELF package, several statistical distributions were fitted to these statistical moments and the distribution with the closest fit to the original percentiles was selected as a prior for modelling. In a few instances, where the best fitting distribution available in SHELF was not supported in GeNIe, the next best fitting distribution was selected as a prior.*

**11) Line 433: You had min, max, and mean values for Koc and half-life. How were these used to discretize?**

We fitted a statistical distribution to these statistical moments in SHELF. Assuming that Koc and half-life come from a truncated normal distribution (truncated at zero), the mean should closely represent the median – hence this was used as the 50th percentile. We also assumed that min closely approximated the 1st percentile and the maximum the 99th percentile and have used them as such. We rounded down/up min and max to the nearest decimal point to make them just a fraction smaller/larger than the 1st and 95th percentiles in the model fitting.

We therefore propose to include this additional explanation in the manuscript as:

*For Koc and hald-life, where we had limited information to inform the prior distribution, we assumed that these variables belonged to a truncated normal distribution (truncated at zero) and we considered the mean to closely represent the median (i.e. the 50th percentile). Further, we assumed that min closely approximated the 1st percentile and the maximum the 99th percentile and have used them as such. We rounded down/up min and max to the nearest decimal point to make them just a fraction smaller/larger than the 1st and 95th percentiles in the model fitting.*

**12) Line 439: the statement "with the number of states maximised to allow sensitivity to change" is not clear. How was sensitivity to change assessed? How did you guarantee that you did not end up with states with no or few cases?**

We propose to clarify this point by adding the following text to the end of this paragraph:

*This was done empirically by progressively increasing the number of discretisation intervals and adjusting discretisation boundaries whilst checking that probabilities of pesticide application rates conditional on Application Change Rate resulted in a probability density shift by one state.*

**13) Please describe how the soils pans were distributed spatially in the fields?**
We assumed a uniform 80% probability of soil pan being present across the catchment as we didn't have any empirical evidence to stratify the probability between different soil hydrological units, crop type or any other spatial aspects.

**14) Line 509: There is a need to define what you mean by maximum intervention.**
We have added the maximum intervention scenario to the list of scenarios in section 2.6 and explained that this is the combination of 50% reduction in pesticide application rate, management of plough pan, delayed application timing and field buffer installation.

**15) Figure 3: How was the half life and Koc nodes populated? You mentioned that you had mean, min and max value only. How were they combined when you had multiple active ingredients?**

Please see the response to comment 11 above. The probability distribution shown for e.g. half-life and Koc in Fig. 3 is the marginal distribution based on the conditional distributions for all active ingredients. However, for simulations of individual pesticide risk, the individual prior conditional distributions are used.

**16) Figure 3: The graphs in the figure are not legible. Please also add units to the x-axis.**
We have increased the font size in Fig. 3 and rearranged the layout to maximise the use of available space. We included units in the node caption as unfortunately the software does not allow us to format the labels on the x- and y- axes. We believe that the current layout is an improvement on the former small-scale image and hope that this will make the figure sufficiently clear.

**17) In the figures please define the names of AI in the captions.**
Thank you for spotting this. This was done for Fig 5 and 6 but not Fig 4. We will change the caption for figure 4 to include AI names.

**18) The transition from the non-spatial GeNIe 3.0 model to the spatial bnspatial model need to explained more. I am assuming that the GeNIe model assumes each field to be a case in the BN and thus the fields are interchangeable. How do these translate when you move to the bnspatial model?**

Please see the response to question 3 above that may help to clarify some of these points. In the spatial implementation of the model, the evidence for each field is provided as a GIS layer. So in effect, the hybrid model simulates average catchment conditions, without taking regard of spatial correlation between variables, whilst the spatial model explicitly models each field using the specific hard evidence available for that location.

In line 476-7 of the revised manuscript, we have added some extra explanation regarding the application of the model spatially as follows:

Available spatial GIS layers were used as 'hard' evidence to set states for relevant nodes in the discretised model and produce spatially explicit simulations of probabilistic outcomes. *Essentially, the discretised model was applied field by field using field-specific inputs from the GIS layers as hard evidence*.

**References**

Bereswill, R., Streloke, M. and Schulz, R.: Risk mitigation measures for diffuse pesticide entry into aquatic ecosystems: proposal of a guide to identify appropriate measures on a catchment scale., Integr. Environ. Assess. Manag., 10(2), 286–298, doi:10.1002/ieam.1517, 2014.

Cash, D., Clark, W. C., Alcock, F., Dickson, N., Eckley, N. and J�ger, J.: Salience, Credibility, Legitimacy and Boundaries: Linking Research, Assessment and Decision Making, SSRN Electron. J., doi:10.2139/ssrn.372280, 2005.

Drewry, J. J., Newham, L. T. H. and Greene, R. S. B.: Index models to evaluate the risk of phosphorus and nitrogen loss at catchment scales, J. Environ. Manage., 92(3), doi:10.1016/j.jenvman.2010.10.001, 2011

Gagkas, Z., Lilly, A., Stutter, A., Baggaley, N. 2019, Development of framework for a Red-Amber-Green assessment on phosphorus application to land. Report prepared for SEPA

Ghahramani, A., Freebairn, D. M., Sena, D. R., Cutajar, J. L. and Silburn, D. M.: A pragmatic parameterisation and calibration approach to model hydrology and water quality of agricultural landscapes and catchments, Environ. Model. Softw., 130(November 2018), 104733, doi:10.1016/j.envsoft.2020.104733, 2020

Grauso, S., Pasanisi, F., Tebano, C., 2018. Assessment of a Simplified Connectivity Index and Specific Sediment Potential in River Basins by Means of Geomorphometric Tools. Geosciences 8, 48, https://doi.org/10.3390/geosciences8020048

Köhne, J. M., Köhne, S. and Šimůnek, J.: A review of model applications for structured soils: b) Pesticide transport, J. Contam. Hydrol., 104(1–4), doi:10.1016/j.jconhyd.2008.10.003, 2009.

Piffady, J., Carluer, N., Gouy, V., le Henaff, G., Tormos, T., Bougon, N., Adoir, E. and Mellac, K.: ARPEGES: A Bayesian Belief Network to Assess the Risk of Pesticide Contamination for the River Network of France, Integr. Environ. Assess. Manag., 0–1, doi:10.1002/ieam.4343, 2020.

Quaglia, G., Joris, I., Broekx, S., Desmet, N., Koopmans, K., Vandaele, K. and Seuntjens, P.: A spatial approach to identify priority areas for pesticide pollution mitigation, J. Environ. Manage., 246(December 2018), 583–593, doi:10.1016/j.jenvman.2019.04.120, 2019.

Reichenberger, S., Bach, M., Skitschak, A. and Frede, H. G.: Mitigation strategies to reduce pesticide inputs into ground- and surface water and their effectiveness; A review, Sci. Total Environ., 384(1–3), 1–35, doi:10.1016/j.scitotenv.2007.04.046, 2007.

Tang, X., Zhu, B. and Katou, H.: A review of rapid transport of pesticides from sloping farmland to surface waters: Processes and mitigation strategies, J. Environ. Sci., 24(3), 351–361, doi:10.1016/S1001-0742(11)60753-5, 2012.

Villamizar, M. L., Stoate, C., Biggs, J., Morris, C., Szczur, J. and Brown, C. D.: Comparison of technical and systems-based approaches to managing pesticide contamination in surface water catchments, J. Environ. Manage., 260, doi:10.1016/j.jenvman.2019.110027, 2020.